# Few-shot Image Generation via Adaptation-Aware Kernel Modulation

**Yunqing Zhao**\*
yunqing_zhao@mymail.sutd.edu.sg

**Keshigeyan Chandrasegaran**\*
keshigeyan@sutd.edu.sg

**Milad Abdollahzadeh**\*
milad_abdollahzadeh@sutd.edu.sg

**Ngai-Man Cheung**†
ngaiman_cheung@sutd.edu.sg

Singapore University of Technology and Design (SUTD)

## Abstract

Few-shot image generation (FSIG) aims to learn to generate new and diverse samples given an extremely limited number of samples from a domain, *e.g.*, 10 training samples. Recent work has addressed the problem using transfer learning approach, leveraging a GAN pretrained on a large-scale source domain dataset and adapting that model to the target domain based on very limited target domain samples. Central to recent FSIG methods are *knowledge preserving criteria*, which aim to select a subset of source model's knowledge to be preserved into the adapted model. However, a *major limitation* of existing methods is that their knowledge preserving criteria consider *only source domain/source task*, and they fail to consider *target domain/adaptation task* in selecting source model's knowledge, casting doubt on their suitability for setups of different *proximity* between source and target domain. **Our work** makes two contributions. As our first contribution, we revisit recent FSIG works and their experiments. Our important finding is that, under setups which assumption of close proximity between source and target domains is relaxed, existing state-of-the-art (SOTA) methods which consider only source domain in knowledge preserving perform *no better* than a baseline fine-tuning method. To address the limitation of existing methods, as our second contribution, we propose *Adaptation-Aware kernel Modulation* (AdAM) to address general FSIG of different source-target domain proximity. Extensive experimental results show that the proposed method consistently achieves SOTA performance across source/target domains of different proximity, including challenging setups when source and target domains are more apart. Project Page: https://yunqing-me.github.io/AdAM/

## 1 Introduction

Generative Adversarial Networks (GANs) [1, 2, 3] have been applied to a range of important applications including image generation [4, 3, 5], image-to-image translation [6, 7], image editing [8, 9], anomaly detection [10], and data augmentation [11, 12]. However, a critical issue is that these GANs often require large-scale datasets and computationally expensive resources to achieve good performance. For example, StyleGAN [4] is trained on Flickr-Faces-HQ (FFHQ) [4] that contains 70,000 images. However, in many practical applications only a few samples are available (*e.g.*, photos of rare animal species / skin diseases). Training a generative model is problematic in this low-data regime, where the generator often suffers from mode collapse or blurred generated images [13, 14, 15]. To address this, *few-shot image generation* (FSIG) studies the possibility of generating

---

\*Equal Contribution      †Corresponding Author

36th Conference on Neural Information Processing Systems (NeurIPS 2022).

Table 1: Transfer learning for few-shot image generation: Various criteria are proposed to *augment* baseline transfer learning to preserve subset of source model's knowledge into the adapted model.

| Method | Knowledge preserving criteria | Source domain/task aware | Target domain/adaptation aware |
|---|---|---|---|
| TGAN [16] | Not available | – | – |
| FreezeD [17] | Preservation of lower layers of the discriminator pre-trained on the *source* domain. | ✓ | ✗ |
| EWC [18] | Preservation of weights important to the *source* generative model pre-trained on the *source* domain. | ✓ | ✗ |
| CDC [14] | Preservation of pairwise distances of generated images by the *source* generative model pre-trained on the *source* domain. | ✓ | ✗ |
| DCL [19] | Preservation of multilevel semantic diversity of the generated images by the *source* generative model pre-trained on the *source* domain. | ✓ | ✗ |
| **AdAM (Our work)** | Preservation of kernels important in ***adaptation*** of source model to *target*. | ✓ | ✓ |

sufficiently diverse and high quality images, given very limited training data (*e.g.*, 10 samples). FSIG also attracts an increasing interest for some downstream tasks, *e.g.*, few-shot classification [12].

**FSIG with Transfer Learning.** Recent works in FSIG are based on transfer learning approach [20] *i.e.*, leveraging the prior knowledge of a GAN pretrained on a large-scale, diverse source dataset (*e.g.*, FFHQ [4] or ImageNet [21]) and adapting it to a target domain with very limited samples (*e.g.*, face paintings [22]). As only very limited samples are provided to define the underlying distribution, standard fine-tuning of a pre-trained GAN suffers from mode collapse: the adapted model can only generate samples closely resembling the given few shot target samples [16, 14]. Therefore, recent works [18, 14, 19] have proposed to *augment* standard fine-tuning with different criteria to carefully preserve subset of source model's knowledge into the adapted model. Various criteria has been proposed (Table 1), and these *knowledge preserving criteria* have been central in recent FSIG research. In general, these criteria aim to preserve subset of source model's knowledge which is deemed to be useful for target-domain sample generation, *e.g.*, improving the diversity of target sample generation.

**Research Gaps.** One major limitation of existing methods is that they consider *only* source domain in preserving subset of source model's knowledge into the adapted model. In particular, these methods *fail to consider* target domain/adaptation task in selection of source model's knowledge (Table 1). For example, EWC [18] applies Fisher Information [23] to select important weights entirely based on the pretrained *source* model, and it aims to preserve these selected weights regardless of the target domain in adaptation. Similar to EWC [18], CDC [14] proposes an additional constraint to preserve pairwise distances of generated images by the *source* model, and there is no consideration of target domain/adaptation. These *target/adaptation-agnostic* knowledge preserving criteria in recent works raise question regarding their suitability in different source/target domain setups. It should be noted that existing FSIG works (under very limited target samples) focus largely on setups where source and target domains are in *close proximity* (semantically) *e.g.*, Human faces (FFHQ)→Baby faces [14, 19], or Cars→Abandoned Cars [14, 19]. It is unclear about their performance when source/target domains are more apart (*e.g.*, Human faces (FFHQ) → Animal faces [5]).

**Contributions.** In this paper we take an important step to address these research gaps for FSIG. Specifically, our work makes two contributions. **As our first contribution**, we revisit existing state-of-the-art (SOTA) algorithms and their experiments. Importantly, we observe that when the close proximity assumption is relaxed in experiment setups and source/target domains are more apart, existing SOTA methods perform *no better* than a baseline fine-tuning method. Our observation suggests that recent methods considering only source domain/source task in knowledge preserving may not be suitable for *general* FSIG when source and target domains are more apart. To validate our claims, we introduce additional experiments with different source/target domains, analyze their proximity qualitatively and quantitatively, and examine existing methods under a unified framework.

Informed by our analysis, **as our second contribution**, we propose an *adaptation-aware kernel modulation* approach to address general FSIG of different source/target domain proximity. In marked contrast to existing works which preserve knowledge important to *source* task, our method aims to preserve subset of source model's knowledge that are important to the *target* domain and the *adaptation* task. More specifically, we propose an *importance probing* algorithm to identify kernels which encode important knowledge for adaptation to the target domain. Then, we preserve the knowledge of these kernels using a parameter-efficient *rank-constrained kernel modulation*.

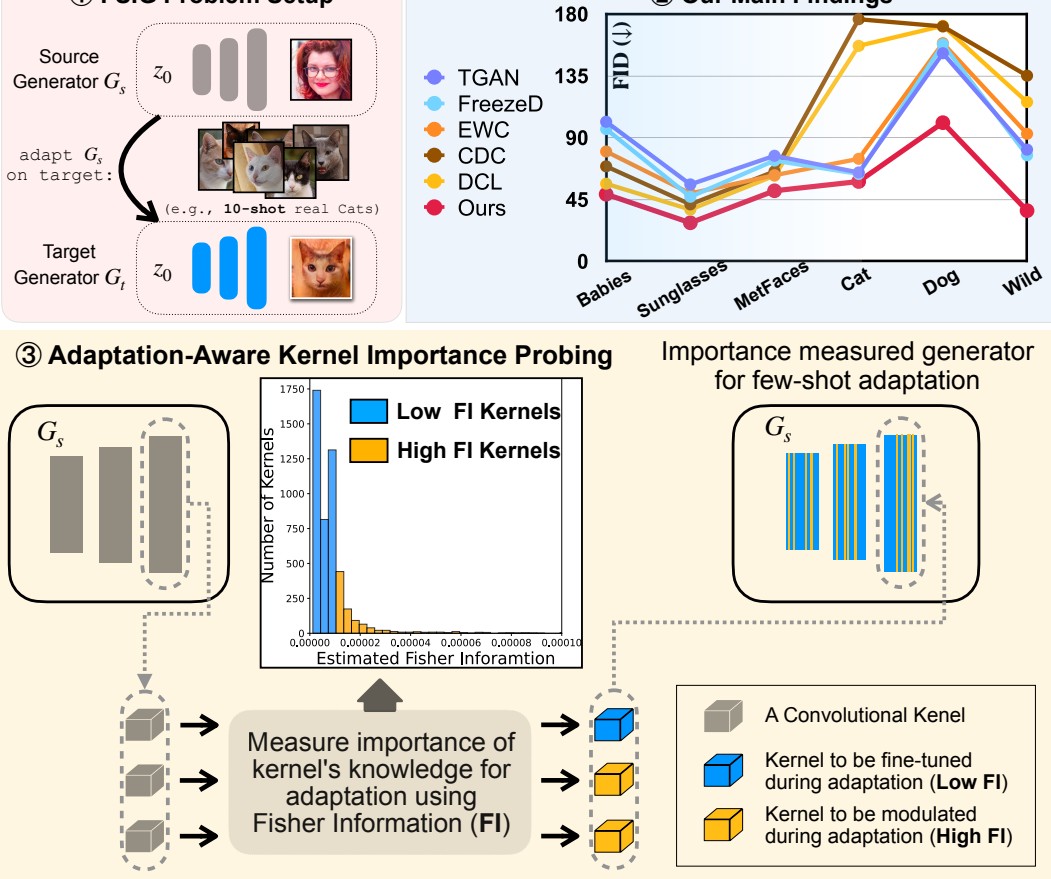

Figure 1: *Overview and our contributions.* ①: We consider the problem of FSIG with Transfer Learning using very limited target samples (*i.e.* 10-shot). ②: Our work makes two contributions, • We discover that when the close proximity assumption between source-target domain is relaxed, SOTA FSIG methods (EWC [18], CDC [14], DCL [19]) which consider only source domain/source task in knowledge preserving perform no better than a baseline fine-tuning method (TGAN [16]) (Sec 3). • We propose a novel adaptation-aware kernel modulation for FSIG that achieves SOTA performance across source / target domains with different proximity (Sec 4). ③ Schematic diagram of our proposed Importance Probing Mechanism: We measure the importance of each kernel for the target domain after probing and preserve source domain knowledge that is important for target domain adaptation (Sec 4). The same operations are applied to discriminator.

We conduct extensive experiments to show that our proposed method consistently achieves SOTA performance across source/target domains of different proximity, including challenging setups when source/target domains are more apart. Our main contributions are summarized as follows:

- We revisit existing FSIG methods and experiment setups. Our study uncovers issues with existing methods when applied to source/target domains of different proximity.
- We propose Adaptation-Aware kernel Modulation (AdAM) for FSIG. Our method consistently achieves SOTA performance both visually and quantitatively across source/target domains with different proximity.

## 2 Related Work

**Few-shot image generation.** Conventional few-shot learning [24, 25, 26] aims at learning a discriminative classifier for classification [27, 28, 29, 30], segmentation [31, 32] or detection [33, 34, 35] tasks. Differently, few-shot image generation (FSIG) [14, 18, 19] aims at learning a generator for new and diverse samples given extremely limited samples (*e.g.*, 10 shots). Transfer learning has been applied to FSIG. For example, Transferring GAN [16] (**TGAN**) applies simple GAN loss [1] to

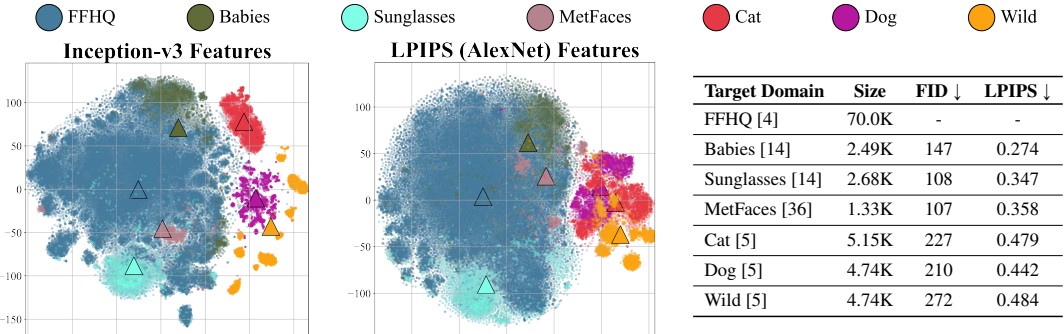

Figure 2: *Qualitative / Quantitative analysis of source-target domain proximity:* We use FFHQ [3] as the source domain. We show source-target domain proximity qualitatively by visualizing Inception-v3 **(Left)** [37] and LPIPS **(Middle)** [38] – using AlexNet [39] backbone – features, and quantitatively using FID / LPIPS metrics **(Right)**. For feature visualization, we use t-SNE [40] and show centroids (△) for all domains. FID / LPIPS is measured with respect to FFHQ. There are two important observations: ① Common target domains used in existing FSIG works (Babies, Sunglasses, MetFaces) are notably proximal to the source domain (FFHQ). This can be observed from the feature visualization and verified by FID / LPIPS measurements. ② We clearly show using feature visualizations and FID / LPIPS measurements that additional setups – Cat [5], Dog [5] and Wild [5] – represent target domains that are distant from the source domain (FFHQ). We remark that large FID values in this analysis are reasonable due to the distance between the source (FFHQ) and different target domains as observed from centroid distance / feature variance. The effect of limited sample size (target domains) for FID / LPIPS measurements are minimal and we include rich supportive studies in Supplementary. Additional experiments and source/target setups in Supplementary to further support our analysis.

fine-tune all parameters of both the generator and the discriminator. **FreezeD** [17] fixes a few high-resolution discriminator layers during fine-tuning. To augment and improve simple fine-tuning, more recent works have focused on preserving specific knowledge from the source models. Elastic weight consolidation (**EWC**) [18] identifies important weights for the *source* model and tries to preserve these weights. Cross-domain Correspondence (**CDC**) [14] preserves pair-wise distance of generated images from the source model to alleviate mode collapse. Dual Contrastive Learning (**DCL**) [19] applies mutual information maximization to preserve multi-level diversity of the generated images by the source model. In this work, we observe that these SOTA methods perform poorly when source and target domains are more apart. Therefore, their proposed source knowledge preservation criteria *may not* be generalizable. Based on our analysis, we propose an adaptation-aware knowledge selection which is more *generalizable* for source/target domains with different proximity.

## 3 Revisiting FSIG through the Lens of Source–Target Domain Proximity

In this section, we revisit existing FSIG methods (10-shot) [16, 17, 18, 14, 19] through the lens of source-target domain proximity. Specifically, we scrutinize the experimental setups of existing FSIG methods and observe that SOTA [18, 14, 19] largely focus on adapting to target domains that are (semantically) proximal to the source domain: Human Faces (FFHQ) → Baby Faces; Human Faces (FFHQ) → Sunglasses; Cars → Abandoned Cars; Church → Haunted Houses [18, 14, 19]. This raises the question as to whether existing source-target domain setups sufficiently represent *general* FSIG scenarios. Particularly, real-world FSIG applications may not contain target domains that are always proximal to the source domain (*e.g.,*: Human Faces (FFHQ) → Animal Faces). Motivated by this, we conduct an in-depth qualitative and quantitative analysis on source-target domain proximity where we introduce target domains that are distant from the source domain (Sec 3.1). Our analysis uncovers an important finding: **Under our additional setups where the assumption of close proximity between source and target domain is relaxed, existing SOTA FSIG methods [18, 14, 19] which consider only source domain/source task in knowledge preserving perform *no better* than a baseline fine-tuning method.** We show this is due to the strong focus of existing SOTA methods in preserving source domain knowledge, thereby not being able to adapt well to distant target domains (Sec 3.2) .

## 3.1 Source–Target Domain Proximity Analysis

**Introducing target domains with varying degrees of proximity to the source domain.** In this section, we formally introduce source-target domain proximity with in-depth analysis to scrutinize existing FSIG methods under different degrees of source-target domain proximity. Following prior FSIG works [16, 17, 18, 14, 19], we use FFHQ [3] as the source domain in this analysis. We remark that existing works largely consider different types of human faces as target domain (*i.e.*: Babies [14], Sunglasses [14], MetFaces [36]), To relax the close proximity assumption and study *general* FSIG problems, we introduce more distant target domains namely Cat, Dog and Wild (from AFHQ [5], consisting of 15,000 high-quality animal face images at $512 \times 512$ resolution) for our analysis.

**Characterizing source-target domain proximity.** Given the wide success of deep neural network features in representing meaningful semantic concepts [41, 42, 43], we visualize Inception-v3 [37] and LPIPS [38] features for source and target domains to qualitatively characterize domain proximity. Further, we use FID [44] and LPIPS distance to quantitatively characterize source-target domain proximity. We remark that FID involves distribution estimation (first, second order moments) [44] and LPIPS computes pairwise distances (learned embeddings) [38] between source / target domains.

**Analysis.** Feature visualization and FID/ LPIPS measurement results are shown in Figure 2. Our results both qualitatively (columns 1, 2) and quantitatively (column 3) show that target domains used in existing works (Babies [3], Sunglasses [3], MetFaces [36]) are notably proximal to the source domain (FFHQ), and our additionally introduced target domains (Dog, Cat and Wild [5]) are distant from the source domain thereby relaxing the close proximity assumption in existing FSIG works.

## 3.2 FSIG methods under Relaxation of Close Domain Proximity Assumption

Motivated by our analysis in Section 3.1, we investigate the performance of existing FSIG methods [16, 17, 18, 14, 19] by relaxing the close proximity assumption between source and target domains. We investigate the performance of these FSIG methods across target domains of different proximity to the source domain, which includes our additionally introduced target domains: Dog, Cat and Wild. The FID results for FFHQ → Cat are: TGAN (simple fine-tuning) [16]: 64.68, EWC [18]: 74.61, CDC [14]: 176.21, DCL [19]: 156.82. Full results can be found in Table 2.

We emphasize that our investigation uncovers an important finding: *Under setups which the assumption of close proximity between source and target domain is relaxed (Dog, Cat, Wild), existing SOTA FSIG methods [18, 14, 19]* perform *no better* than a baseline method [16]. This can be consistently observed in Table 2.

This finding is critical as it exposes a serious drawback of SOTA FSIG methods [18, 14, 19] when close domain proximity (between source and target) assumption is relaxed. We further analyse generated images from SOTA FSIG methods and observe that these methods are unable to adapt well to distant target domains due to *only considering source domain / task in knowledge preservation.* This can be clearly observed from Figure 3. We remark that TGAN (simple baseline) [16] also suffers from severe mode collapse. Given that our investigation uncovers an important problem in SOTA FSIG methods, we tackle this problem in Sec 4. Figure 3 (last row) shows a glimpse of our proposed method.

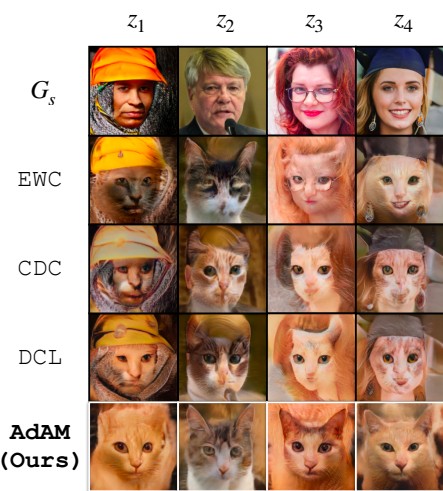

Figure 3: $G_s$ is the source generator (FFHQ). Adapting from the source domain (FFHQ) to a distant target domain (Cat) using SOTA FSIG methods EWC [18], CDC [14], DCL [19] (rows 2, 3, 4) results in observable knowledge transfer that is not useful to the target domain. *i.e.*: Source task knowledge such as *Caps ($z_1, z_4$), Hair styles/color – brown ($z_2$), red-hair ($z_3$), Eye glasses ($z_3$)* from FFHQ are transferred to Cats during adaptation which is not appropriate. Our method (last row) can alleviate these issues.

## 4 Adaptation-Aware Kernel Modulation

We focus on this question: *"Given a pretrained GAN on a source domain $\mathcal{D}_s$, and a few-samples from a target domain $\mathcal{D}_t$, which part of the source model's knowledge should be preserved, and which part should be updated, during the adaptation from $\mathcal{D}_s$ to $\mathcal{D}_t$?"* In contrast to

**Algorithm 1:** Few-Shot Image Generation via Adaptation-Aware Kernel Modulation (AdAM)

**Require:** Pre-trained GAN: $G_s$ and $D_s$, $iter_{probe}$, $iter_{adapt}$, threshold quantile $t$, learning rate $\alpha$

*Importance Probing:*

1   Freeze all kernels $\{\mathbf{W}_i\}_{i=1}^N$ in pre-trained networks $G_s$, and $D_s$

2   Randomly initialize a modulation matrix $\mathbf{M}_i$ for each kernel $\mathbf{W}_i$

3   **for** $k = 0$, $k{+}{+}$, *while* $k < iter_{probe}$ **do**

4      Perform kernel modulation for all kernels using Eqn.1 to obtain modulated weights $\hat{\mathbf{W}}$

5      Update $\mathbf{M} \leftarrow \mathbf{M} - \alpha \nabla_{\mathbf{M}} \mathcal{L}(G(z); \hat{\mathbf{W}})$ `/*` `lightweight, i.e.,` $iter_{probe} << iter_{adapt}$ `*/`

6   **end**

7   Measure importance of each kernel $\mathbf{W}_i$ by computing FI for the corresponding $\mathbf{M}_i$ using Eqn.3

8   Compute the index set $\mathcal{A}$ of important kernels using quantile $t$ of FI values as threshold

*Main Adaptation:*

9   **if** $j \in \mathcal{A}$ **then**

10      Initialize the kernel by $\mathbf{W}_j$ and freeze the kernel, randomly initialize $\mathbf{M}_j$

11   **else**

12      Initialize the kernel by $\mathbf{W}_j$

13   **end**

14   **for** $k = 0$, $k{+}{+}$, *while* $k < iter_{adapt}$ **do**

15      **if** $j \in \mathcal{A}$ **then**

16          Modulate kernel using Eqn.1 to obtain modulated weights $\hat{\mathbf{W}}_j$

17          Update $\mathbf{M}_j \leftarrow \mathbf{M}_j - \alpha \nabla_{\mathbf{M}_j} \mathcal{L}(G(z); \hat{\mathbf{W}})$

18      **else**

19          Update $\mathbf{W}_j \leftarrow \mathbf{W}_j - \alpha \nabla_{\mathbf{W}_j} \mathcal{L}(G(z); \hat{\mathbf{W}})$

20      **end**

21   **end**

SOTA FSIG methods [18, 14, 19], we propose an adaptation-aware FSIG that also considers the target domain / adaptation task in deciding which part of the source model's knowledge to be preserved. In a CNN, each *kernel* is responsible for a specific part of knowledge (*e.g.*, pattern or texture). Similar behaviour is also observed for both generator [45] and discriminator [46] in GANs. Therefore, in this work, we make this knowledge preservation decision at the kernel level, *i.e.*, ***casting the knowledge preservation to a decision problem of whether a kernel is important when adapting from $\mathcal{D}_s$ to $\mathcal{D}_t$.***

Our FSIG algorithm has two main steps: (i) a lightweight *importance probing* step, and (ii) *main adaptation* step. In the first step, *i.e.*, importance probing, we adapt the model using a parameter-efficient design to the target domain for a limited number of iterations, and during this adaptation, we measure the importance of each individual kernel for the *target domain*. The output of importance probing are decisions of importance / unimportance of individual kernels. Then, in the second step, *i.e.*, main adaptation, we preserve the knowledge of important kernels and update the knowledge of unimportant kernels. The overview of the proposed system is shown in Figure 1 and the pseudocode is shown in Algorithm 1.

**Proposed Importance Probing for FSIG.** Our intuition for the proposed importance probing is: *"The source GAN kernels have different levels of importance for each target domain."* For example, different subsets of kernels could be important when adapting a pretrained GAN on FFHQ to Babies [14] compared to adapting the same pretrained GAN to Cat [5]. Therefore, we aim for a knowledge preservation criterion that is target domain/adaptation-aware (Table 1). In order to achieve adaptation-awareness, we propose a light-weight importance probing algorithm which considers adaptation from source to target domain. There are two important design considerations: probing under (i) extremely limited number of target data and (ii) low computation overhead.

As discussed, in this *importance probing* step, we adapt the source model to the target domain for a limited number of iterations and with a few available target samples. During this short adaptation step, we measure the importance of kernel for the adaptation task. To measure the importance, we use Fisher information (FI) which gives the *informative knowledge* of that kernel in handling adaptation task [47]. Then, based on FI measurement, we classify kernels into important / unimportant. These kernel-level importance decisions are then used in the next step, *i.e.*, main adaptation.

In the main adaptation step, we propose to apply *kernel modulation* to achieve restrained update for the important kernels, and *simple fine-tuning* for the unimportant kernels. As will be discussed, the modulation is rank-constrained and has restricted degree-of-freedom; therefore, it is capable to preserve knowledge of the important kernels. On the other hand, simple fine-tuning has large degree-of-freedom for updating knowledge of the unimportant kernels. Furthermore, the rank-constrained kernel modulation is parameter-efficient. Therefore, we also apply this rank-constrained kernel modulation *in the probing step* to determine the importance of kernels.

**Kernel Modulation.** The kernel modulation is used in the main adaptation step to preserve knowledge of important kernels into the adapted model. Furthermore, it is also used in the probing step as a parameter-efficient technique to determine importance of kernels. Specifically, we apply Kernel ModuLation (KML) which is proposed very recently [29]. In [29], KML is proposed for multimodal few-shot *classification* (FSC). In particular, in [29], KML has been found to be effective for knowledge transfer between different *classification* tasks of different modes under few-shot constraint. Therefore, in our work, we apply KML for knowledge transfer between different *generation* tasks of different domains under limited target domain samples.

Specifically, in each convolutional layer of a CNN, the $i^{th}$ kernel of that layer $\mathbf{W}_i \in \mathbb{R}^{c_{in} \times k \times k}$ is convolved with the input feature $\mathbf{X} \in \mathbb{R}^{c_{in} \times h \times w}$ to the layer to produce the $i^{th}$ output channel (feature map) $\mathbf{Y}_i \in \mathbb{R}^{h' \times w'}$, *i.e.*, $\mathbf{Y}_i = \mathbf{W}_i * \mathbf{X} + b_i$, where $b_i \in \mathbb{R}$ denotes the bias term. Then, KML [29] modulates $\mathbf{W}_i$ by multiplying it with the modulation matrix $\mathbf{M}_i \in \mathbb{R}^{c_{in} \times k \times k}$ plus an all-ones matrix $\mathbf{J} \in \mathbb{R}^{c_{in} \times k \times k}$:

$$\hat{\mathbf{W}}_i = \mathbf{W}_i \odot (\mathbf{J} + \mathbf{M}_i) \tag{1}$$

where $\odot$ denotes Hadamard multiplication. In Eqn. 1, using $\mathbf{J}$ allows to learn the modulation matrix in a residual format. Therefore, the modulation weights are learned as perturbations around the pretrained kernels which helps to preserve source knowledge. The exact pretrained kernel can also be transferred to the target model if it is optimal. There are some important differences between discriminative version of KML in [29] and our version, please see Supplementary for details.

This baseline KML learns an individual modulation parameter for each coefficient of the kernel. Therefore, it could suffer from *parameter explosion* when using in recent GAN architectures (*e.g.*, more than 58M parameters in StyleGAN-V2 [3] [1]). To address this issue, instead of learning the modulation matrix, we learn a *low-rank* version of it [29, 48]. More specifically, for a Conv layer within CNN, with a total number of $d_{out}$ kernels to be modulated, instead of learning $\mathbf{M} = \{\mathbf{M}_i\}_{i=1}^{d_{out}}$, we learn two proxy vectors $\mathbf{m}_1 \in \mathbb{R}^{d_{out}}$, and $\mathbf{m}_2 \in \mathbb{R}^{(c_{in} \times k \times k)}$, and construct the modulation matrix using the outer product of these vectors, *i.e.*, $\mathbf{M} = \mathbf{m}_1 \otimes \mathbf{m}_2$. Furthermore, as we are using KML for adaptable knowledge preservation, we *freeze* the base kernel $\mathbf{W}_i$ during adaptation. Therefore, trainable parameters are $\mathbf{m}_1, \mathbf{m}_2$. This reduces the number of trainable parameters significantly, and has better performance on restraining the update of important kernels (see Supplementary). As it will be discussed later, the value of $d_{out}$ equals to the total number of kernels in a layer ($c_{out}$) during probing, and for main adaptation, it is determined by the output of our probing method ($d_{out} \leqslant c_{out}$).

**Importance Measurement.** Recall our FSIG has two main steps: (i) importance probing step (Lines 1-8 in Algorithm 1), and (ii) main adaptation step (Lines 9-21 in Algorithm 1). In probing, we also apply KML as a parameter-efficient technique to determine importance of individual kernels. In particular, for probing, we propose to apply KML to all kernels (in both generator and discriminator) to identify which of the *modulated* kernels are important for the adaptation task. To measure the importance of the modulated kernels, we apply Fisher information (FI) to the modulation parameters. In our FSIG setup, for a modulated GAN with parameters $\Theta$, Fisher information $\mathcal{F}$ can be computed as:

$$\mathcal{F}(\Theta) = \mathbb{E}\big[ -\frac{\partial^2}{\partial \Theta^2} \mathcal{L}(x|\Theta) \big] \tag{2}$$

where $\mathcal{L}(x|\Theta)$ is the binary cross-entropy loss computed using the output of the discriminator, and $x$ includes few-shot target samples, and fake samples generated by GAN. Then, FI for a modulation matrix $\mathcal{F}(\mathbf{M}_i)$ can be computed by averaging over FI values of parameters within that matrix. As we are using the low-rank estimation to construct the modulation matrix, we can estimate $\mathcal{F}(\mathbf{M}_i)$ by FI values of the proxy vectors. In particular, considering the outer product in low-rank approximation, we have $\mathbf{M}_i = reshape([\mathbf{m}_1^i \mathbf{m}_2^1, \ldots, \mathbf{m}_1^i \mathbf{m}_2^{(c_{in} \times k \times k)}])$, where $|\mathbf{m}_2| = c_{in} \times k \times k$. Then we

[1] https://github.com/rosinality/stylegan2-pytorch

Table 2: FSIG (10-shot) results: We report FID scores (↓) of our proposed *adaptation-aware* FSIG and compare with existing FSIG methods. We emphasize that Cat, Dog and Wild target domains are additional experiments included in this work. (Sec 3.1). Our experiment results show two important findings: **1)** Under setups which assumption of close proximity between source and target domains is relaxed (Cat, Dog, Wild), SOTA FSIG methods – EWC, CDC, DCL – which consider only source domain in knowledge preserving perform *no better* than a baseline fine-tuning method (TGAN). **2)** Our proposed adaptation-aware FSIG achieves SOTA performance in *all* target domains due to preserving source domain knowledge that is important for few-shot target domain adaptation. We generate 5,000 images using the adapted generator to evaluate FID on the whole target domain. We also report the corresponding KID, Intra-LPIPS and standard deviations in Supplementary.

| Target Domain | Babies [14] | Sunglasses [14] | MetFaces [36] | AFHQ-Cat [5] | AFHQ-Dog [5] | AFHQ-Wild [5] |
|---|---|---|---|---|---|---|
| TGAN [16] | 101.58 | 55.97 | 76.81 | 64.68 | 151.46 | 81.30 |
| TGAN+ADA [36] | 97.91 | 53.64 | 75.82 | 80.16 | 162.63 | 81.55 |
| FreezeD [17] | 96.25 | 46.95 | 73.33 | 63.60 | 157.98 | 77.18 |
| EWC [18] | 79.93 | 49.41 | 62.67 | 74.61 | 158.78 | 92.83 |
| CDC [14] | 69.13 | 41.45 | 65.45 | 176.21 | 170.95 | 135.13 |
| DCL [19] | 56.48 | 37.66 | 62.35 | 156.82 | 171.42 | 115.93 |
| **AdAM (Ours)** | **48.83** | **28.03** | **51.34** | **58.07** | **100.91** | **36.87** |

use the unweighted average of FI for parameters of $\mathbf{m}_1$ and $\mathbf{m}_2$, proportional to their occurrence frequency in calculation of $\mathbf{M}_i$, as an estimate of $\mathcal{F}(\mathbf{M}_i)$ (details in Supplementary):

$$\hat{\mathcal{F}}(\mathbf{M}_i) = \mathcal{F}(\mathbf{m}_1^i) + \frac{1}{|\mathbf{m}_2|} \sum_{j=1}^{|\mathbf{m}_2|} \mathcal{F}(\mathbf{m}_2^j) \tag{3}$$

After calculating $\hat{\mathcal{F}}(\mathbf{M}_i)$ for all modulation matrices in both generator and discriminator, we use the $t\%$ quantile of these values as a threshold (separately for generator and discriminator) to decide whether modulation of a kernel is important or unimportant for adaptation to the target domain. If the modulation of a kernel is determined to be important (during probing), the kernel is modulated using KML during main adaptation step; otherwise, the kernel is updated using simple fine-tuning during main adaptation. In all setups, we perform probing for 500 iterations. We remark that in probing only modulation parameters $\mathbf{m}_1, \mathbf{m}_2$ are trainable, and FI is only computed on them, therefore the probing is a very lightweight step and can be performed with minimal overhead (details in Supplementary). The output of probing step are the decisions to apply kernel modulation or simple fine-tuning on individual kernels. Then, based on these decisions, the main adaptation is performed. The proposed FSIG scheme is summarized in Algorithm 1.

## 5 Empirical Studies

### 5.1 Experiments / Results

**Experiment Details.** For fair comparison, we strictly follow prior works [16, 17, 18, 14, 19] in the choice of GAN architecture, source-target adaptation setups and hyper-parameters. We use StyleGAN-V2 [3] as the GAN architecture and FFHQ as the source domain. Our experiments include setups with different source-target proximity: Babies/Sunglasses [14], MetFaces [36] and Cat/Dog/Wild (AFHQ) [5] (See Sec. 3). Adaptation is performed with 256 x 256 resolution and batch size 4 on a single Tesla V100 GPU. We apply importance probing and modulation on base kernels of both generator and discriminator. We focus on 10-shot target adaptation setup in the main paper.

**Qualitative Results.** We show generated images with our proposed AdAM along Baseline [16, 17] and SOTA FSIG methods [18, 14, 19] for two target domains, Babies and Cat with different degrees of proximity to FFHQ, before and after adaptation. The results are shown in Figure 4 top and bottom, respectively. By preserving source domain knowledge that is important for target domain, our proposed adaptation-aware FSIG method can generate substantially high quality images with high diversity for both Babies and Cat domains. We also include FID [44] and Intra-LPIPS [14] (for measuring diversity) to quantitatively show that our proposed method outperforms SOTA FSIG methods [18, 14, 19]. We show more generated samples in Supplementary.

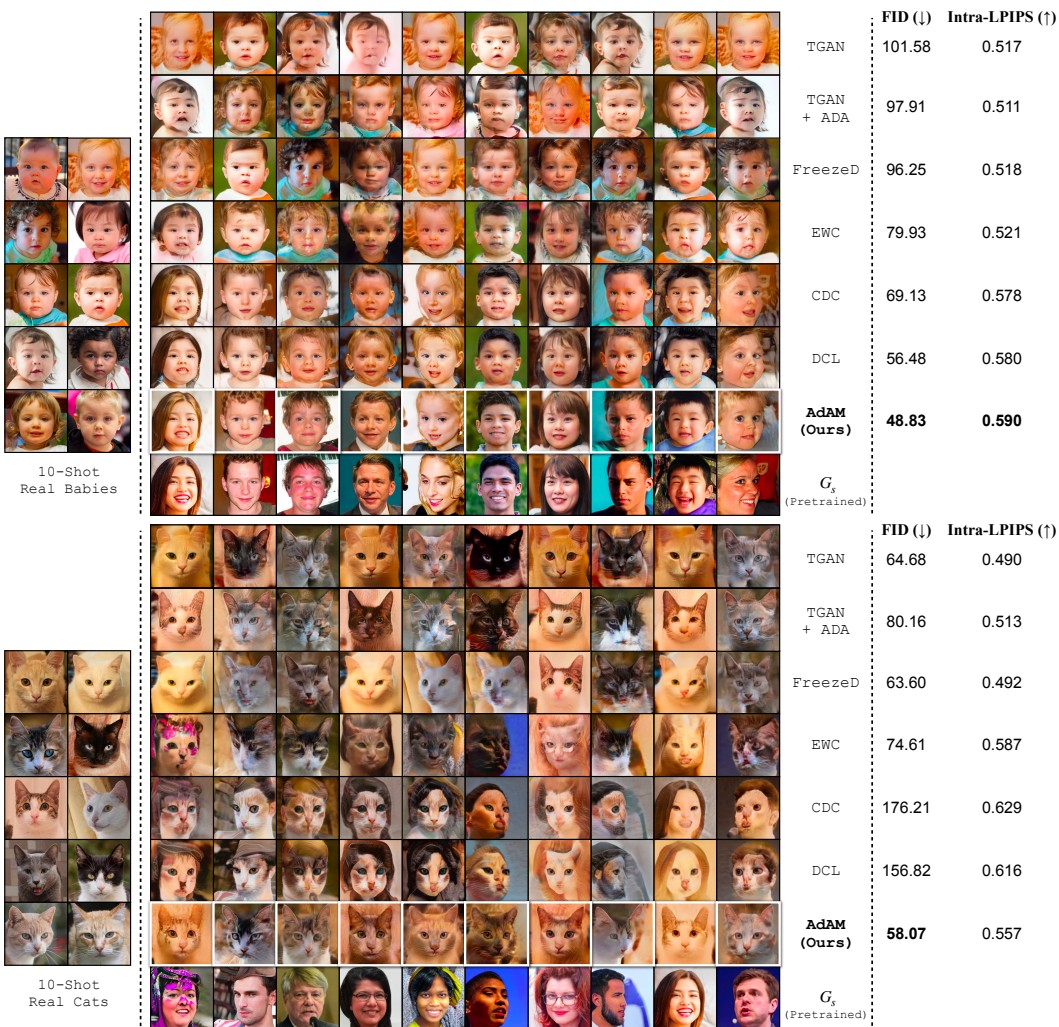

Figure 4: Qualitative and quantitative comparison of 10-shot image generation with different FSIG methods. Images of each column are from the same noise input. **Left**: 10 real target images for few-shot adaptation. **Middle, Right**: For target domain with close proximity (*e.g.*Babies, top), our method can generate high quality images with more refined details and diversity knowledge, achieving best FID and Intra-LPIPS socre. For target domain which is distant (*e.g.*, Cat, bottom), TGAN/FreezeD overfit to the 10-shot samples and others fail. In contrast, our method preserves meaningful semantic features at different levels (*e.g.*, posture and color) from source, achieving a good trade off between quality and diversity. In particular, our Intra-LPIPS approaches that of EWC, while our generated images have much better quality qualitatively and quantitatively.

**Quantitative Results.** We show complete FID (↓) scores in Table 2. Our proposed AdAM for FSIG achieves SOTA results across all target domains of varying proximity to the source (FFHQ). We emphasize that it is achieved by preserving source domain knowledge that is important for target domain adaptation (Sec 4). We also report Intra-LPIPS (↑) as an indicator of diversity, as Figure 4.

## 5.2 Analysis

**Ablation study of Importance Probing.** The goal of importance probing (denoted as "IP") is to identify kernels that are important for *few-shot target adaptation* as shown in Figure 5 (Top). To justify the effectiveness of our design choice, we perform an ablation study that discards the IP stage and regard all kernels as *equally important* for target adaptation. Therefore, we simply modulate all kernels *without any knowledge selection*. As one can observe from Figure 5 (Bottom), knowledge selection plays a vital role in adaptation performance. Specifically, the significance of knowledge preservation is more evident when the target domains are distant from the source domain.

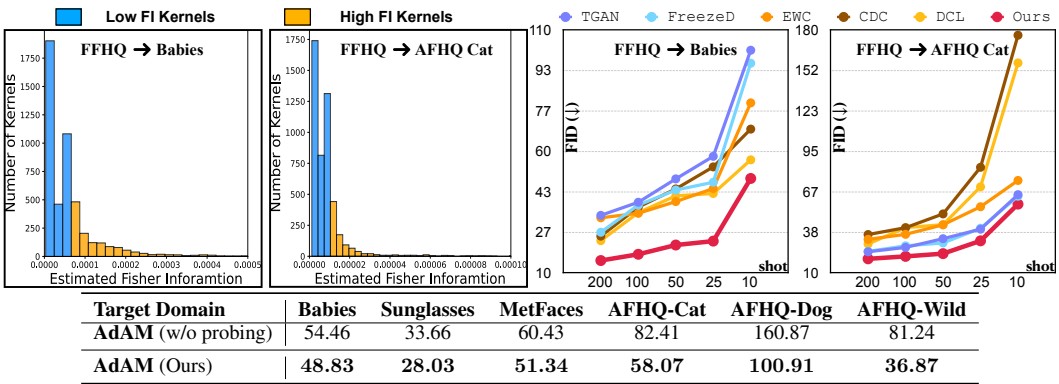

Figure 5: **(Top Left)** Our proposed IP identifies and preserves source kernels important (high FI) for target adaptation. **(Bottom)** FID score on different datasets. We validate the effectiveness of IP by modulating all kernels without IP. On the other hand, if we fine-tune all parameters without IP and modulation (TGAN), it suffers mode collapse (Table 2 and Figure 4). **(Top Right)** We evaluate the performance of different number of shots (10, 25, 50, 100, 200) on Babies and AFHQ-Cat. We show that our method consistently outperforms other FSIG methods in all setups. In Supplementary, we also show the generated images given different number of shots on more target domains.

**Number of target samples (shots).** The number of target domain training samples is an important factor that can impact the FSIG performance. In general, more target domain samples can allow better estimation of target distribution. We study the efficacy of our proposed method under different number of target domain samples. The results are shown in Figure 5, and we show that our proposed adaptation-aware FSIG method consistently outperforms existing methods in all setups.

## 6  Discussion

**Conclusion.** Focusing on FSIG, we make two contributions. First, we revisit current SOTA methods and their experiments. We discover that SOTA methods perform poorly in setups when source and target domains are more distant, as existing methods only consider source domain/task for knowledge preservation. Second, we propose a new FSIG method which is target/adaptation-aware (AdAM). Our proposed method outperforms previous work across all setups of different source-target domain proximity. We include extended experiments and analysis in Supplementary.

**Broader Impact.** Our work makes contribution to generation of synthetic data in applications where sample collection is challenging, *e.g.*, photos of rare animal species. This is an important contribution to many data-centric applications. Furthermore, transfer learning of generative models using a few data sample enables data and computation-efficient model development. Our work has positive impact on environmental sustainability and reduction of greenhouse gas emission. While our work targets generative applications with limited-data, it parallely raises concerns regarding such methods being used for malicious purposes. Given the recent success of forensic detectors [49, 50, 51, 52], we conduct a simple study using Color-Robust forensic detector proposed in [49] on our Babies and Cat datasets. We observe that the model achieves 99.8% and 99.9% average precision (AP) respectively showing that AdAM samples can be successfully detected. We also remark that our work presents opportunities for improving knowledge transfer methods [53, 54, 55, 56] in a broader context.

**Limitations.** While our experiments are extensive compared to previous works, in practical applications, there are many possible target domains which cannot be included in our experiments. However, as our method is target/adaptation aware, we believe our method can generalize better than existing SOTA which are target-agnostic.

## Acknowledgment

This research is supported by the National Research Foundation, Singapore under its AI Singapore Programmes (AISG Award No.: AISG2-RP-2021-021; AISG Award No.: AISG-100E2018-005). This project is also supported by SUTD project PIE-SGP-AI-2018-01. We thank anonymous reviewers for their insightful comments.

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
