# Few-shot Image Generation via Adaptation-Aware Kernel Modulation — Supplementary —

## Overview

This Supplementary provides additional experiments, results, analysis and ablation studies to further support our contributions. The Supplementary materials are organized as follows:

**Reproducibility.** Code and pre-trained models: https://yunqing-me.github.io/AdAM/.

Table S1: Comparison of training cost in terms of number of trainable parameters, training iterations and compute time for different FSIG methods. FFHQ is the source domain and we show results for Babies (top) and Cat (bottom) target domains. One can clearly observe that our proposed IP is extremely lightweight and our KML based adaptation contains much less trainable parameters in the source GAN. All results are measured in containerized environments using a single Tesla V100-PCIE (32 GB) GPU with batch size of 4. All reported results are averaged over 3 independent runs.

**FFHQ → Babies**

| Method | Stage | # trainable params (M) | # iteration | # time |
|---|---|---|---|---|
| TGAN [1] | Adaptation | 30.0 | 3000 | 110 mins |
| FreezeD [2] | Adaptation | 30.0 | 3000 | 110 mins |
| EWC [3] | Adaptation | 30.0 | 3000 | 110 mins |
| CDC [4] | Adaptation | 30.0 | 3000 | 120 mins |
| DCL [5] | Adaptation | 30.0 | 3000 | 120 mins |
| **AdAM (Ours)** | IP | **0.105** | **500** | 8 mins |
| | Adaptation | **18.9** | **1500** | 65min |

**FFHQ → AFHQ-Cat**

| Method | Stage | # trainable params (M) | # iteration | # time |
|---|---|---|---|---|
| TGAN [1] | Adaptation | 30.0 | 6000 | 210 mins |
| FreezeD [2] | Adaptation | 30.0 | 6000 | 200 mins |
| EWC [3] | Adaptation | 30.0 | 6000 | 220 mins |
| CDC [4] | Adaptation | 30.0 | 6000 | 300 mins |
| DCL [5] | Adaptation | 30.0 | 6000 | 300 mins |
| **AdAM (Ours)** | IP | **0.105** | **500** | 8 mins |
| | Adaptation | **18.9** | **2500** | 110 mins |

## A    Proposed Importance Probing Algorithm: Details

### A.1    Computational Overhead

Our proposed Importance Probing (IP) algorithm to measure the importance of each individual kernel in the source GAN for the target-domain is lightweight. *i.e.*: proposed importance probing only requires 8 minutes compared to the adaptation step which requires $\approx$ 110 minutes (Averaged over 3 runs for FFHQ → Cat adaptation experiment). This is achieved using two design choices:

- During IP, only modulation parameters are updated. Given that our modulation design is low-rank KML, the number of trainable parameters is significantly small compared to the actual source GAN. *i.e.*: number of trainable parameters in our proposed IP is only 0.1M whereas the source GAN contains 30.0M trainable parameters.

- Our proposed IP is performed for limited number of iterations to measure the importance for the target domain. *i.e.*: IP stage requires only 500 iterations to achieve a good performance for adaptation.

Complete details on number of trainable parameters and compute time for our proposed method and existing FSIG works are provided in Table S1. As one can observe, our proposed method (both IP and adaptation) is better than existing FSIG works in terms of trainable parameters and compute time.

### A.2    Kernel Modulation (KML) with rank-constrained operations

Here we show more details of KML, as supplement to the main paper, as Figure S1.

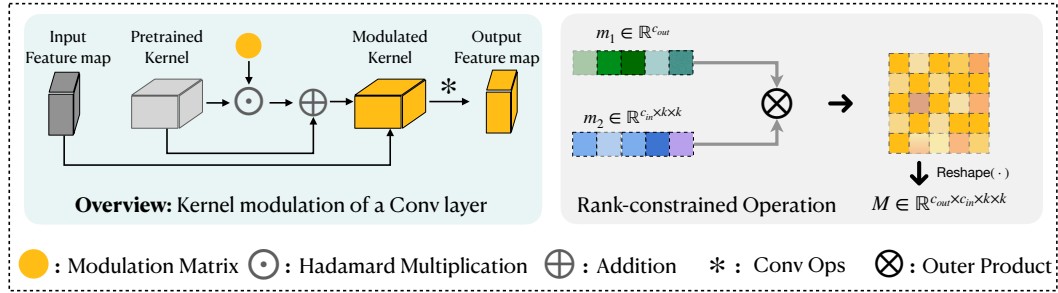

Figure S1: Illustration of Kernel Modulation operations. Here we use a convolutional kernel for instance. Similar operations are applied to the linear layer.

## A.3 Fisher Information Approximation Using Proxy Vectors

Recall in Sec.4 of main paper, we consider low-rank approximation of modulation matrix using outer product of proxy vectors: $\mathbf{M}_i = reshape([\mathbf{m}_1^i \mathbf{m}_2^1, \ldots, \mathbf{m}_1^i \mathbf{m}_2^{c_{in} \times k \times k}])$, where $|\mathbf{m}_2| = (c_{in} \times k \times k)$. In order to calculate the FI of the modulation matrix, we start with the FI of each element in this matrix. Considering $m_{ij} = \mathbf{m}_1^i \mathbf{m}_2^j$, following equation can be derived by simple application of chain rule of differentiation:

$$\frac{\partial \mathcal{L}}{\partial m_{ij}} = \frac{1}{2\mathbf{m}_2^j}\frac{\partial \mathcal{L}}{\partial \mathbf{m}_1^i} + \frac{1}{2\mathbf{m}_1^i}\frac{\partial \mathcal{L}}{\partial \mathbf{m}_2^j} \tag{1}$$

We use the square of the gradients to estimate the FI [6]. Therefore, the following equation can be obtained between the FI of these variables:

$$\mathcal{F}(m_{ij}) = \frac{1}{4\mathbf{m}_2^{j^2}}\mathcal{F}(\mathbf{m}_1^i) + \frac{1}{4\mathbf{m}_1^{i^2}}\mathcal{F}(\mathbf{m}_2^j) + \frac{1}{2\mathbf{m}_1^i \mathbf{m}_2^j}\frac{\partial \mathcal{L}}{\partial \mathbf{m}_1^i}\frac{\partial \mathcal{L}}{\partial \mathbf{m}_2^j} \tag{2}$$

Then, the FI of the modulation matrix $\mathbf{M}_i = [m_{i1}, m_{i2}, \ldots]$, can be calculated as:

$$
\begin{aligned}
\mathcal{F}(\mathbf{M}_i) &= \sum_{j=1}^{|\mathbf{m}_2|}\mathcal{F}(m_{ij}) \\
&= \sum_{j=1}^{|\mathbf{m}_2|}\left(\frac{1}{4\mathbf{m}_2^{j^2}}\mathcal{F}(\mathbf{m}_1^i) + \frac{1}{4\mathbf{m}_1^{i^2}}\mathcal{F}(\mathbf{m}_2^j) + \frac{1}{2\mathbf{m}_1^i \mathbf{m}_2^j}\frac{\partial \mathcal{L}}{\partial \mathbf{m}_1^i}\frac{\partial \mathcal{L}}{\partial \mathbf{m}_2^j}\right) \\
&= \mathcal{F}(\mathbf{m}_1^i)\sum_{j=1}^{|\mathbf{m}_2|}\frac{1}{4\mathbf{m}_2^{j^2}} + \frac{1}{4\mathbf{m}_1^{i^2}}\sum_{j=1}^{|\mathbf{m}_2|}\mathcal{F}(\mathbf{m}_2^j) \\
&\quad + \frac{1}{2\mathbf{m}_1^i}\frac{\partial \mathcal{L}}{\partial \mathbf{m}_1^i}\sum_{j=1}^{|\mathbf{m}_2|}\frac{1}{\mathbf{m}_2^j}\frac{\partial \mathcal{L}}{\partial \mathbf{m}_2^j}
\end{aligned}
\tag{3}
$$

We empirically observed that discarding (i) the cross-term (ii) the coefficients ($\frac{1}{4\mathbf{m}_2^{j^2}}$, $\frac{1}{4\mathbf{m}_1^{i^2}}$) in the importance of each kernel in Eqn. 3 results in a similar FID for the final adapted model. Therefore, the estimation can be simpler and more lightweight. In particular, the following (simpler) estimated version of $\mathcal{F}(\mathbf{M}_i)$ is used in our work:

$$\hat{\mathcal{F}}(\mathbf{M}_i) = \mathcal{F}(\mathbf{m}_1^i) + \frac{1}{|\mathbf{m}_2|}\sum_{j=1}^{|\mathbf{m}_2|}\mathcal{F}(\mathbf{m}_2^j) \tag{4}$$

Note that $\hat{\mathcal{F}}(\mathbf{M}_i)$ intuitively estimates the FI of the modulation matrix by a weighted average of its constructing parameters corresponding to their occurrence frequency in calculation of $\mathbf{M}_i$. We remark that in our implementation, for reporting all of the results in the main paper, and also the additional results in the supplementary, we have used this lightweight estimation Eqn. 4 to calculate the importance of each kernel during importance probing.

## B    Discussion of Related Works

In Sec.2 of the main paper, we discuss closely-related work of this paper that focuses on few-shot image generation (FSIG) under extremely limited data, i.e., 10 samples. Here, we review other related work.

### B.1    Image generation with less data

Since the introduction of GANs [7], there is a fair amount of work to focus on training of GANs with less data in recent literature, with efforts on introducing additional data augmentation methods [8, 9], regularization terms [10], modifying GAN architectures [11], and modification of filter kernels [11, 12]. Commonly, these works focus on setups with several thousands of images, i.e.: Flowers dataset [13] with 8,189 images in [12], 10% of ImageNet, or the entire AFHQ [14] dataset. On the other hand, FSIG with extremely limited data (10 samples) poses unique challenges. In particular, as pointed out in [4, 5], severe mode collapse and loss in diversity are critical challenges in FSIG that require special attention. We remark that in [12], a technique called AdaFM is introduced to update kernels. However, the underlying ideas and mechanism of AdaFM and our KML are quite different. AdaFM is inspired from style-transfer literature [15], introduces independent scale and shift (scalar) parameters to update individual channels of kernels to manipulate their styles. On the other hand, as discussed in the main paper, KML introduces a structural $\mathbf{J} + \mathbf{M}$, $\mathbf{M} = \mathbf{m}_1 \otimes \mathbf{m}_2$, to update multiple kernels in a coordinated manner. In our experiment, we also test AdaFM in few-shot setups and compare its performance with KML.

### B.2    Discriminative kernel modulation

As mentioned in the main paper, Kernel ModuLation (KML) is originally proposed in [16] for adapting the model between different modes of few-shot classification (FSC) tasks. However, due to some differences between the multimodal meta-learner in [16], and our transfer learning-based scheme, there are important differences in design choices when applying KML to our problem. ***First***, in contrast to FSC work [16] which follows a *discriminative learning* setup, we aim to address a problem in a *generative learning* setup. ***Second***, in FSC setup, the modulation parameters are generated during adaptation to target task with a pretrained modulation network trained on tens of thousands of few-shot tasks. So the modulation parameters are not directly learned for a target few-shot task. In contrast, in our setup, the base kernel is frozen during the adaptation, and we directly learn the modulation parameters for a target domain/task using a very limited number of samples (e.g., 10-shot). ***Finally***, in FSC, usually source and target tasks follow a same task distribution $p(\mathcal{T})$. In fact, in implementation, even though the classes are disjoint between source and target tasks, all of them are constructed using the data from the same domain (e.g., miniImageNet [17]). However, in our setup, the source and target tasks/domains distributions could be very different (e.g., Human Faces (FFHQ) → Cats).

## C    Ablation Studies and Additional Analysis on Importance Probing

In this section, we conduct extensive ablation studies to show the significance of our proposed method for FSIG. Similar to main paper analysis, we use FFHQ [18] as the source domain, and use Babies and Cat [14] as target domains. The different approaches in the study are as follows:

- TGAN [1]: The source GAN models pretrained on FFHQ are updated using *simple fine-tuning* with the 10 shot target samples.
- EWC [3]: Following [3], a L2 regularization is applied to all model weights to augment simple fine-tuning. The regularization is scaled by the importance of individual model weights as determined by the FI of the model weights based on the *source* models.

- EWC + IP: We apply our probing idea on top of EWC. In the probing step, original EWC as discussed above is used but with a small number of iterations. At the end of probing, FI of the model weights based on the *updated* models is computed. Then, during main adaptation, this *target-aware* FI is used to scale the L2 regularization. In other words, EWC + IP is a target-aware version of EWC in [3] using our probing idea.

- AdaFM [12]: AdaFM modulation is applied to all kernels.

- AdaFM + IP: We apply our probing idea on top of AdaFM. In the probing step, original AdaFM as discussed above is used but with a small number of iterations. At the end of probing, FI of AdaFM parameters is computed, and kernels are classified as important/unimportant using the same 75% quantile threshold as in our work. Then, during main adaptation, the important kernels are updated via AdaFM, and the unimportant kernels are updated via simple fine tuning. In other words, AdaFM + IP is a target-aware version of AdaFM using our probing idea.

- Ours w/o IP (*i.e.* main adaptation only): KML modulation is applied to all kernels.

- Ours w/ Freeze: We apply our probing idea as discussed in the main paper, *i.e.*, with KML applied to all kernels but adaptation with a small number of iterations. At the end of probing, FI of KML parameters is computed, and kernels are classified as important/unimportant using the same 75% quantile threshold as in our work. Then, during main adaptation, the important kernels are *frozen*, and the unimportant kernels are updated via simple fine tuning. In other words, this is similar to our proposed method except that kernel freezing is used in main adaptation instead of KML for important kernels.

- Ours w/ KML (*i.e.* our main proposed method): This is the method proposed in the main paper. We apply our probing idea as discussed in the main paper, *i.e.*, with KML applied to all kernels but adaptation with a small number of iterations. At the end of probing, FI of KML parameters is computed, and kernels are classified as important/unimportant using 75% quantile threshold. Then, during main adaptation, the important kernels are modulated using KML, and the unimportant kernels are updated via simple fine tuning.

**Qualitative Results.** We show generated images corresponding to all approaches discussed above in Figure S2. These results show that our proposed idea on importance probing is principally a suitable approach to improve FSIG by identifying kernels important for target domain adaptation. Figure S2 also shows that our proposed method can generate images with better quality.

**Quantitative results.** We show FID / LPIPS results in Table S2. These results show that our proposed IP is principally a suitable approach for FSIG. This can be clearly observed when applying IP to EWC [3] and AdaFM [12]. We remark that probing with KML (ours AdAM) is computationally much efficient compared to probing with EWC and AdaFM due to less number of trainable parameters. Overall, we quantitatively show that our proposed method outperforms existing FSIG methods with IP, thereby generating images with a good balance between quality (FID ↓) and diversity (Intra-LPIPS ↑). We also empirically observe that methods performing IP at kernel level (Ours w/ KML, AdaFM + IP) perform better than method performing IP at parameter level (EWC + IP).

# D   Extended Experiments and Results

In this section, we conduct additional experiments to further support our findings and contributions.

## D.1   Additional source / target domains

Following [4], we conduct extended experiments using Church as the source domain. [4] uses Haunted houses and Van Gogh Houses as target domains. Similar to Sec.3 in the main paper, our analysis confirms that these target domains are closer to the source domain (Church). We additionally include palace and yurt as target domains to relax the close proximity assumption. Proximity visualization is shown in Figure S3.

**Experiment Details.** For fair comparison, we strictly follow prior works [1, 2, 3, 4, 5] in the choice of GAN architecture, source-target adaptation setups and hyper-parameters. We use StyleGAN-V2 [18] as the GAN architecture and FFHQ as the source domain. We use 256 x 256 resolution for adaptation. Adaptation is performed with batch size 4 on a single Tesla V100 GPU. We apply importance probing

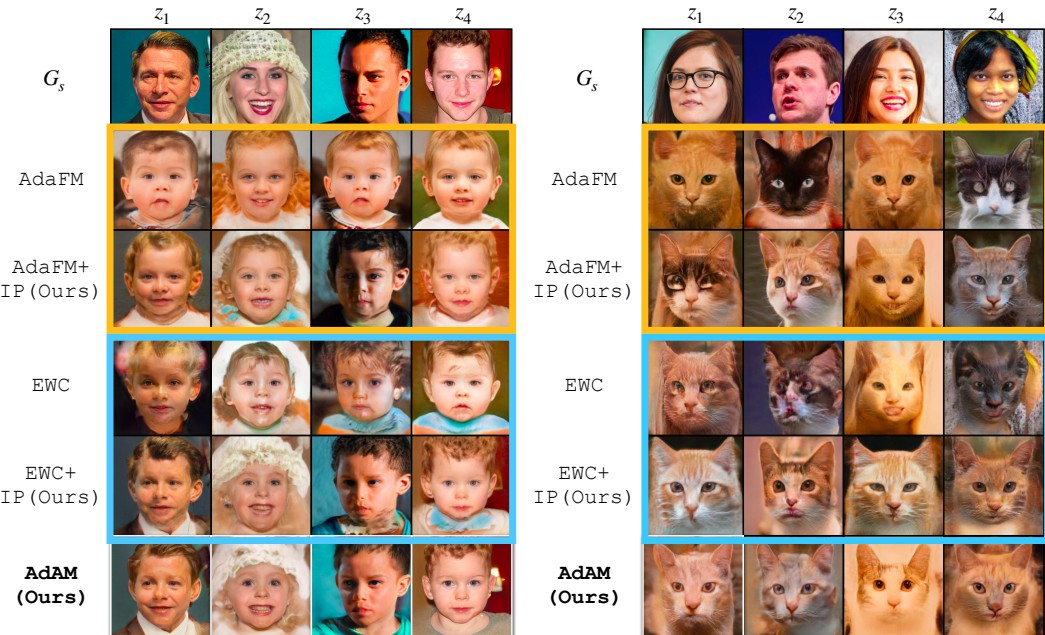

Figure S2: $G_s$ is the source generator (FFHQ). We show results for FFHQ → Babies (left) and FFHQ → Cat (right), similar to the main paper. Applying our idea of importance probing to EWC [3], AdaFM [12], we observe better quality in FSIG. This shows that our proposed idea on importance probing is principally a suitable approach to improve FSIG. One can also observe that images generated by our proposed method (with KML) has good quality compared to other methods. This is quantitatively confirmed in Table S2 .

and modulation on base kernels of both generator and discriminator. We focus on 10-shot target adaptation setup.

**Results.** Given that the target domain only contains 10 real images, following [4], we show the quality of FSIG for 10-shot adaption. Qualitative analysis is shown in Figure S19. As one can observe, SOTA FSIG methods [3, 4, 5] are unable to adapt well to distant target domain (palace) due to *due to only considering source domain / task in knowledge preservation.* We remark that TGAN [1] suffers severe mode collapse. We clearly show that our proposed adaptation-aware FSIG method outperforms existing FSIG works.

Further, we show complete 10-shot adaptation results. Results for Haunted Houses, Van Gogh Houses, Palace and Yurt are shown in Figures S4, S5, S7, S6 respectively. Other adaptation setups with FFHQ/LSUN-Cars as source are shown in Figure S20, Figure S21, Figure S22, Figure S23, Figure S24, Figure S25 and Figure S26.

### D.2 Additional GAN Architectures

We use an additional pre-trained GAN architecture, ProGAN [24], to conduct FSIG experiments for FFHQ → Babies, FFHQ → Cat, Church → Haunted houses and Church → Palace setups. For fair comparison, we strictly follow the exact experiment setup discussed in Section D.1.

**Results.** We show complete qualitative and quantitative results for FFHQ → Babies, FFHQ → Cat adaptation in Figures S8 and S9 respectively. As one can observe, our proposed method consistently outperforms other baseline and SOTA FSIG methods with another pre-trained GAN model (ProGAN [24]), demonstrating the effectiveness and generalizability of our method. We also show qualitative results for Church → Haunted houses and Church → Palace adaptation in Figures S10 and S11 respectively.

Table S2: Ablation studies for IP: FFHQ [18] is the source domain. We use Babies and Cats [14] as target domains. We show FID (left) and Intra-LPIPS (right) results. For each method, best FID and LPIPS results are shown in **bold**. IP is performed for 500 iterations (where relevant). These results show that our proposed IP is principally a suitable approach for FSIG. This can be clearly observed when applying IP to EWC [3] (EWC+IP) and AdaFM [12] (AdaFM+IP). We also observe that methods performing IP at kernel level (Ours w/ KML, AdaFM + IP) perform better than method performing IP at parameter level (EWC + IP). Overall, we quantitatively show that our proposed method outperforms all existing FSIG methods with IP, thereby generating images with high quality (FID) and diversity (Intra-LPIPS).

| Target Domain | Babies | Cat | Target Domain | Babies | Cat |
|---|---|---|---|---|---|
| | FID ($\downarrow$) | | | Intra-LPIPS ($\uparrow$) | |
| TGAN [1] | 101.58 | 64.68 | TGAN [1] | 0.517 | 0.490 |
| EWC [3] | 79.93 | 74.61 | EWC [3] | 0.521 | **0.587** |
| EWC + [IP (Ours)] | **70.80** | **66.35** | EWC + [IP (Ours)] | **0.625** | 0.540 |
| AdaFM [12] | 62.90 | 64.44 | AdaFM [12] | 0.568 | 0.525 |
| AdaFM + [IP (Ours)] | **55.64** | **60.04** | AdaFM + [IP (Ours)] | **0.577** | **0.540** |
| Ours w/o IP | 54.46 | 82.41 | Ours w/o IP | **0.613** | 0.522 |
| Ours w/ Freeze [w/ IP] | 50.81 | 61.60 | Ours w/ Freeze [w/ IP] | 0.581 | 0.559 |
| **AdAM** (w/ KML [w/ IP]) | **48.83** | **58.07** | **AdAM** (w/ KML [w/ IP]) | 0.590 | 0.557 |

## D.3 Alternative characterization of importance measure

In literature, Class Salience [25] (CS) is used as a property to explain which area/pixels of an input image stand out for a specific classification decision. Similar to the estimated Fisher Information (FI) used in our work, the complexity of CS is based on the first-order derivatives. Therefore, conceptually CS could have a connection with FI as they both use the knowledge encoded in the gradients.

We perform an experiment to replace FI with CS in importance probing and compare with our original approach. Note that, in [25], CS is computed w.r.t. input image pixels. To make CS suitable for our problem, we modify it and compute CS w.r.t. modulation parameters. Similar to our approach in the main paper, we average the importance of all parameters within a kernel to calculate the importance of that kernel. Then we use these values during our importance probing to determine the important kernels for adapting from source to target domain (as Sec. 4 in our main paper). The results in Table S3 are obtained with our proposed method using FI and CS during importance probing:

Table S3: In this experiment, we replace FI with CS in importance probing and compare with our original approach. We evaluate the performance under different source $\rightarrow$ target adaptation setups.

| Domain | FFHQ $\rightarrow$ Babies | | FFHQ $\rightarrow$ Cat | |
|---|---|---|---|---|
| | FID ($\downarrow$) | Intra-LPIPS ($\uparrow$) | FID ($\downarrow$) | Intra-LPIPS ($\uparrow$) |
| Class Salience [25] | 52.46 | 0.582 | 61.68 | 0.556 |
| Fisher Information (Ours) | **48.83** | **0.590** | **58.07** | **0.557** |

Our results suggest that importance probing using FI (approximated by first-order derivatives) can perform better in selection of important kernels, leading to better performance (FID, intra-LPIPS) in the adapted models as shown in the Table S3.

## D.4 Comparison with Adaptive Data Augmentation [8]

We additionally include the results of Adaptive Data Augmentation [8] (ADA), as a supplement to Figure 4 in the main paper. We show that our proposed method consistently outperforms ADA in few-shot adaptation setups. The results are shown in Figures S12 and S13.

## D.5 Importance probing with extremely limited number of samples.

In Figure 6 (main paper), we perform ablation studies to show that our method consistently outperforms other baseline and SOTA methods given different number of target samples. In this section, we

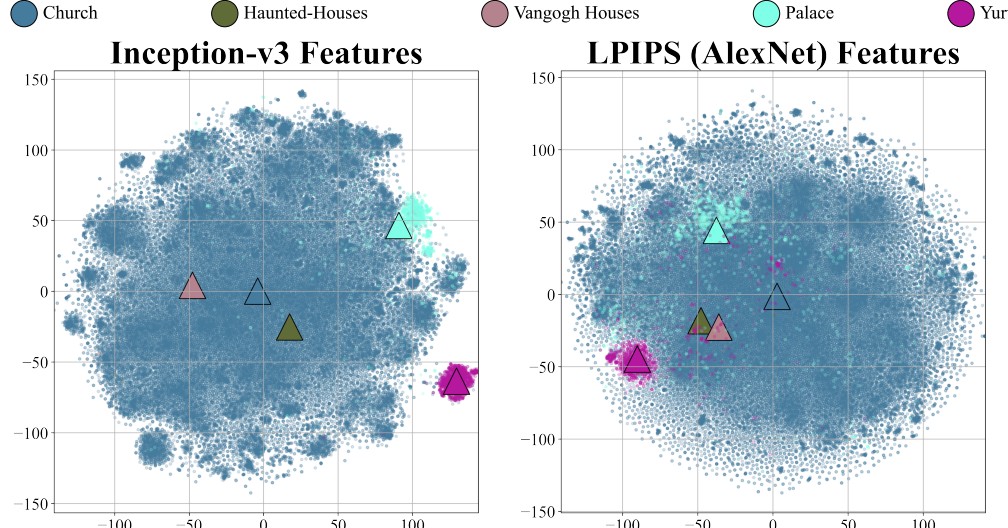

Figure S3: *Source-target domain proximity Visualization:* We use Church as the source domain following [4]. We show source-target domain proximity by visualizing Inception-v3 (Left) [19] and LPIPS (Middle) [20] –using AlexNet [21] backbone– features, and quantitatively using FID / LPIPS metrics (Right). For feature visualization, we use t-SNE [22] and show centroids (△) for all domains. FID / LPIPS is measured with respect to FFHQ. There are 2 important observations: ① Common target domains used in existing FSIG works (Haunted Houses, Van Gogh Houses) are notably proximal to the source domain (Church). This can be observed from the feature visualization and verified by FID / LPIPS measurements. ② We clearly show using feature visualizations and FID / LPIPS measurements that additional setups – Palace [23] and Yurt [23] – represent target domains that are distant from the source domain (Church). We remark that due to availability of only 10-shot samples in the target domain, FID / LPIPS are not measured in these setups.

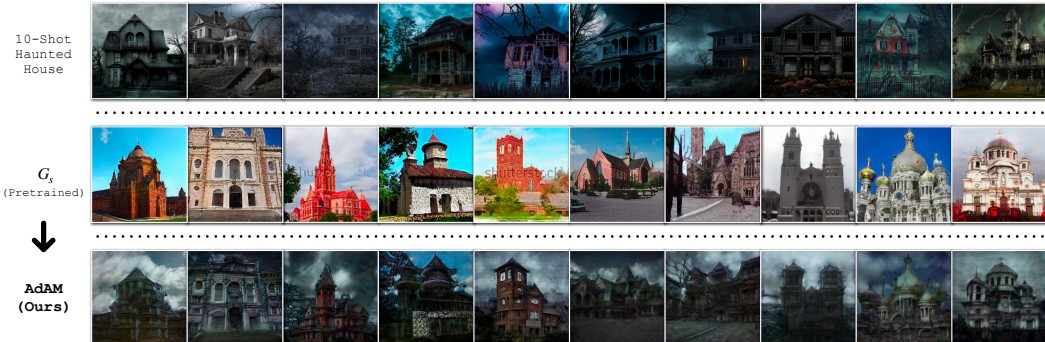

Figure S4: Church → Haunted House

conduct additional experiments with extremely limited number of target samples: 1-shot and 5-shot. We also conduct experiments with more training samples during adaptation to show that our method consistently outperforms existing FSIG methods.

**Results.** The results are shown in Figure S14. Here we additionally include Adaptive Data Augmentation as an important baseline, and qualitative results can be found in Figures S12 and S13.

## E   Discussion: What form of visual information is encoded by high FI kernels?

In this section, we attempt to discover what form of visual information is encoded/generated by a specific high FI kernel identified by our importance probing method. This is a complex problem and to our best knowledge, methods on visualizing generative models/GANs are still rather restrictive in

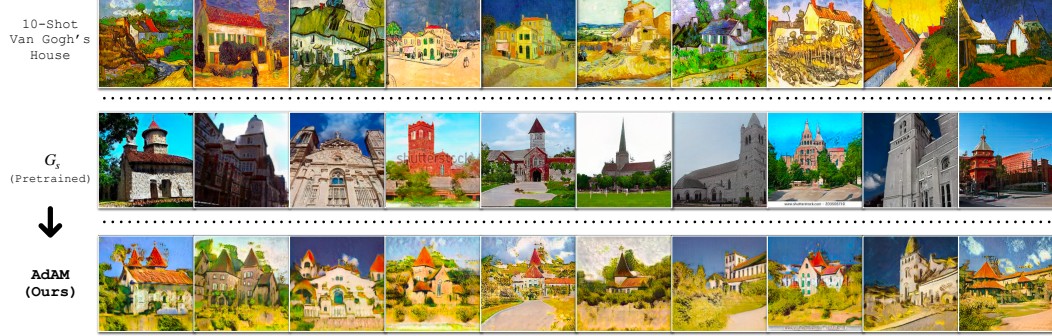

Figure S5: Church → Van Gogh's House

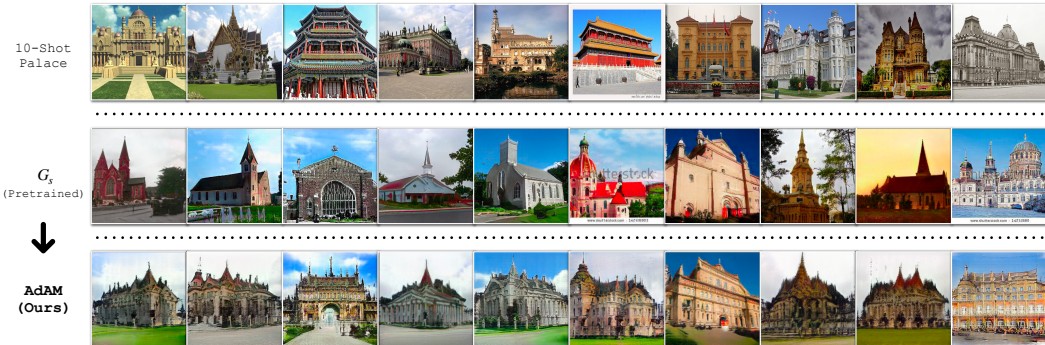

Figure S6: Church → Palace (distant domain)

terms of concepts that can be visualized. Nevertheless, we leverage on GAN Dissection method [26], a more established visualization method to visualize the high FI internal representations.

**Experiment setup:** We use Church as the source domain as official GAN Dissection method [1] is more suitable for scene-based image generation models (This is due to limitation of the semantic segmentation pipeline in GAN Dissection [26]). We use 2 target domains: haunted houses (proximal domain) and palace (distant domain). Following official GAN Dissection implementation [26], we use the ProGAN [24] model. For fair comparison, we strictly follow the exact experiment setup discussed in Section D.1.

**Results.**

- Visualizing high FI kernels for Church → Haunted Houses adaptation : The results for FI estimation for kernels and several distinct semantic concepts learnt by high FI kernels are shown in Figure S15. In Figure S15, we visualize four examples of high FI kernels: (a), (b), (c), (d) corresponding to concepts building, building, tree and wood respectively. Using GAN Dissection, we observe that a notable amount of high FI kernels correspond to useful source domain concepts including building, tree and wood (texture) which are preserved when adapting to Haunted Houses target domain. We remark that these preserved concepts are useful to the target domain for adaptation.

- Visualizing high FI kernels for Church → Palace adaptation : The results for FI estimation for kernels and several distinct semantic concepts learnt by high FI kernels are shown in Figure S16. In Figure S16, we visualize four examples of high FI kernels: (a), (b), (c), (d) corresponding to concepts grass, grass, building and building respectively. Using GAN Dissection, we observe that a notable amount of high FI kernels correspond to useful source domain concepts including grass and building which are preserved when adapting to Palace target domain. We remark that these preserved concepts are useful to the target domain (Palace) for adaptation.

---

[1] https://github.com/CSAILVision/gandissect

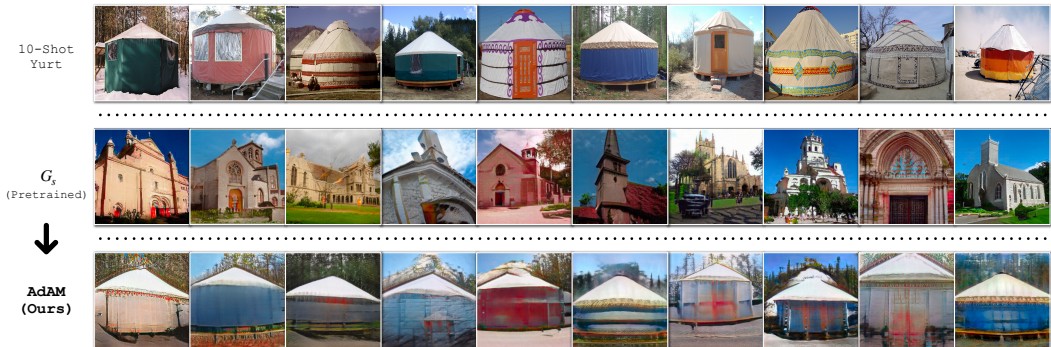

Figure S7: Church → Yurt (distant domain)

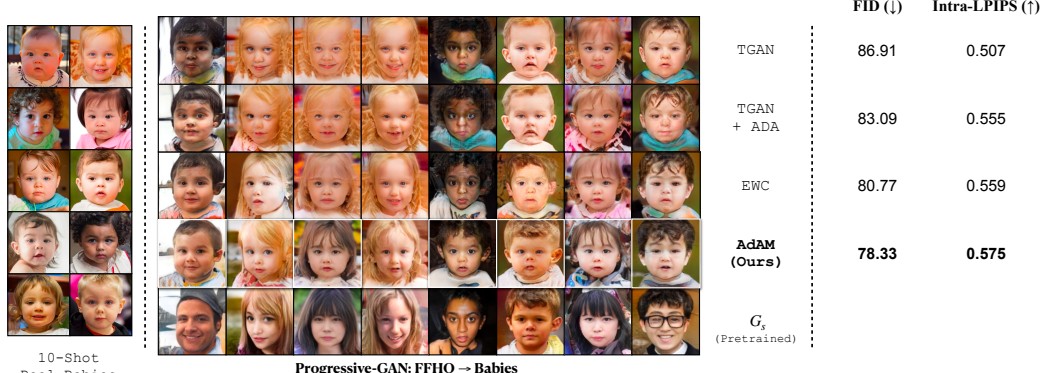

| | FID (↓) | Intra-LPIPS (↑) |
|---|---|---|
| TGAN | 86.91 | 0.507 |
| TGAN + ADA | 83.09 | 0.555 |
| EWC | 80.77 | 0.559 |
| **AdAM (Ours)** | **78.33** | **0.575** |
| $G_s$ (Pretrained) | | |

Figure S8: FFHQ → Babies 10-shot adaptation results using pre-trained ProGAN [24] generator. We include ADA results. As one can observe, our proposed method outperforms existing FSIG methods.

**Limitations of GAN Dissection / Future Work :** Although GAN Dissection can uncover useful semantic concepts preserved by high FI kernels, GAN Dissection method [26] is limited by the dataset used for semantic segmentation. Hence this method is not able to uncover concepts that are not present in semantic segmentation dataset (They use Broaden Dataset [27]). Therefore, using GAN dissection we are currently unable to discover and visualize more fine-grained concepts preserved by our high FI kernels. We hope to further address this problem in future work.

# F   Main Paper Experiments : Additional Results / Analysis

## F.1   KID / Intra-LPIPS / Standard Deviation of Experiments

**KID / Intra-LPIPS.** In addition to FID scores reported in the main paper, we evaluate KID [28] and Intra-LPIPS [20]. We remark the KID (↓) is another metric in addition to FID (↓) to measure the quality of generated samples, and Intra-LPIPS (↑) measures the diversity of generated samples. In literature, the original LPIPS [20] evaluates the perceptual distance between images. We follow CDC [4] and DCL [5] to measure the Intra-LPIPS, a variant of LPIPs, to evaluate the degree of diversity. Firstly, we generate 5,000 images and assign them to one of 10-shot target samples, based on the closet LPIPS distance. Then, we calculate the LPIPS of 10 clusters and take average. KID and Intra-LPIPS results are reported in Tables S4 and S5 respectively. As one can observe, our proposed adaptation-aware FSIG method outperforms SOTA FSIG methods [3, 4, 5] and produces high quality images with good diversity.

**Standard Deviation of FID scores.** We report standard deviation of FID scores for Babies and Cat corresponding to the main paper experiments (Table 2: main paper) in Table S6. As one can observe, the standard deviations are within acceptable range.

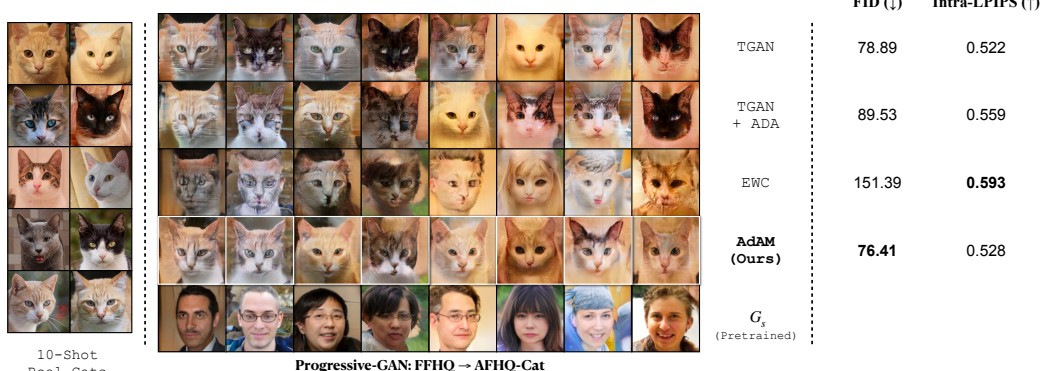

|  | FID (↓) | Intra-LPIPS (↑) |
|---|---|---|
| TGAN | 78.89 | 0.522 |
| TGAN + ADA | 89.53 | 0.559 |
| EWC | 151.39 | **0.593** |
| AdAM (Ours) | **76.41** | 0.528 |
| $G_s$ (Pretrained) | | |

Figure S9: FFHQ → Cat 10-shot adaptation results using pre-trained ProGAN [24] generator. We include ADA results. As one can observe, our proposed method outperforms existing FSIG methods.

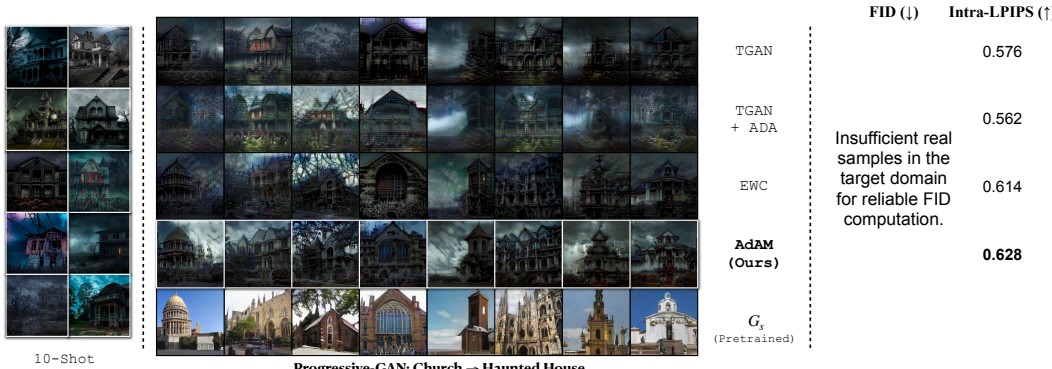

|  | FID (↓) | Intra-LPIPS (↑) |
|---|---|---|
| TGAN | | 0.576 |
| TGAN + ADA | Insufficient real samples in the target domain for reliable FID computation. | 0.562 |
| EWC | | 0.614 |
| AdAM (Ours) | | **0.628** |
| $G_s$ (Pretrained) | | |

Figure S10: Church → Haunted Houses 10-shot adaptation results using pre-trained ProGAN [24] generator. We include ADA results.

Table S4: KID (↓) score of different methods with the same checkpoint of Table 2 in the main paper. The values are in $10^3$ units, following [8, 29].

| Method | TGAN | FreezeD | EWC | CDC | DCL | AdAM (Ours) |
|---|---|---|---|---|---|---|
| Babies | 81.92 | 65.14 | 51.81 | 51.74 | 43.46 | **28.38** |
| AFHQ-Cat | 41.912 | 38.834 | 58.65 | 196.60 | 117.82 | **32.78** |

Table S5: Intra-LPIPS (↑) of different methods, the standard deviation is calculated over 10 clusters. Compared to the baseline models (TGAN/FreezeD) or state-of-the-art FSIG methods (EWC/CDC/DCL), our proposed method can achieve a good trade-off between diversity and quality of the generated images, see Table 2 in main paper for FID score.

| Method | TGAN | FreezeD | EWC | CDC | DCL | AdAM (Ours) |
|---|---|---|---|---|---|---|
| Babies | $0.517 \pm 0.04$ | $0.518 \pm 0.05$ | $0.521 \pm 0.03$ | $0.578 \pm 0.03$ | $0.580 \pm 0.02$ | $0.590 \pm 0.03$ |
| AFHQ-Cat | $0.490 \pm 0.02$ | $0.492 \pm 0.04$ | $0.587 \pm 0.04$ | $0.629 \pm 0.03$ | $0.616 \pm 0.05$ | $0.557 \pm 0.02$ |

Table S6: FID score (↓) with standard deviation over 3 different runs.

| Method | TGAN | FreezeD | EWC | CDC | DCL | AdAM (Ours) |
|---|---|---|---|---|---|---|
| Babies | $101.69 \pm 0.50$ | $97.15 \pm 1.02$ | $79.59 \pm 0.26$ | $66.98 \pm 1.58$ | $56.64 \pm 0.90$ | **$47.92 \pm 0.87$** |
| AFHQ-Cat | $64.60 \pm 0.68$ | $64.56 \pm 0.69$ | $74.69 \pm 0.32$ | $174.5 \pm 2.55$ | $154.60 \pm 1.98$ | **$57.59 \pm 0.36$** |

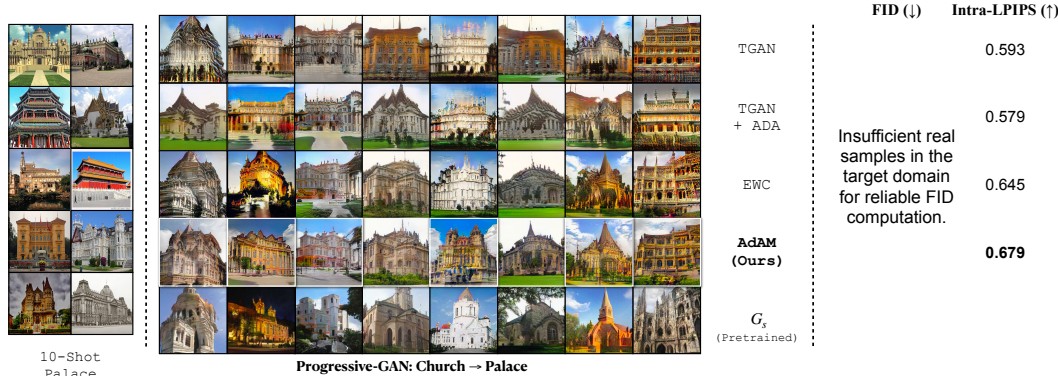

| | FID (↓) | Intra-LPIPS (↑) |
|---|---|---|
| TGAN | | 0.593 |
| TGAN + ADA | Insufficient real samples in the target domain for reliable FID computation. | 0.579 |
| EWC | | 0.645 |
| **AdAM (Ours)** | | **0.679** |
| $G_s$ (Pretrained) | | |

Figure S11: Church → Palace 10-shot adaptation results using pre-trained ProGAN [24] generator. We include ADA results.

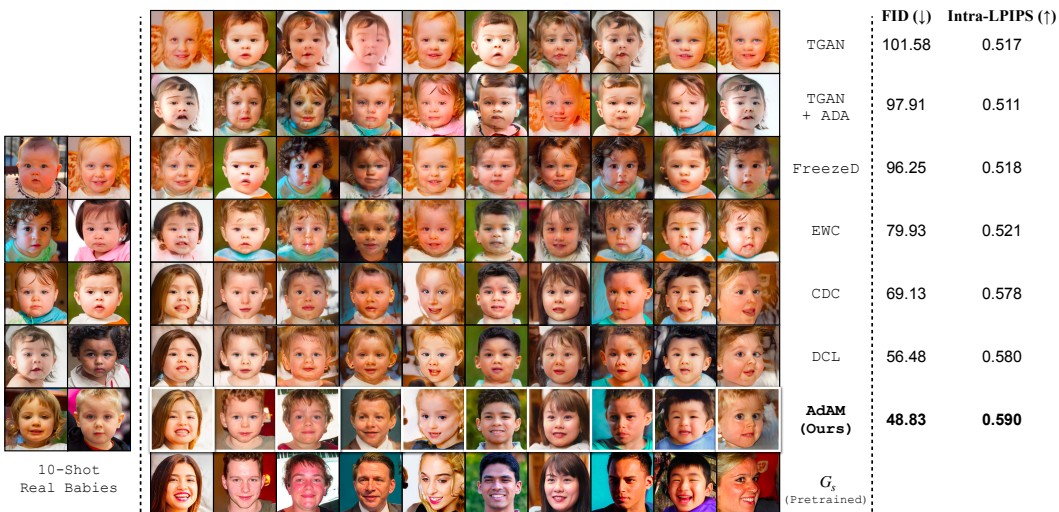

| | FID (↓) | Intra-LPIPS (↑) |
|---|---|---|
| TGAN | 101.58 | 0.517 |
| TGAN + ADA | 97.91 | 0.511 |
| FreezeD | 96.25 | 0.518 |
| EWC | 79.93 | 0.521 |
| CDC | 69.13 | 0.578 |
| DCL | 56.48 | 0.580 |
| **AdAM (Ours)** | **48.83** | **0.590** |
| $G_s$ (Pretrained) | | |

Figure S12: FFHQ → Babies results, including ADA [8].

## F.2 10-shot Adaptation Results

We show complete 10-shot adaptation results for our proposed adaptation-aware FSIG method and existing FSIG methods [1, 3, 4, 5] for distant target domains. Results for FFHQ → Dog and FFHQ → Wild are shown in Figures S17 and S18 respectively. As one can observe, SOTA FSIG methods [3, 4, 5] are unable to adapt well to distant target domains (palace, yurt) due to *due to only considering source domain / task in knowledge preservation.* We remark that TGAN [1] suffers severe mode collapse. We clearly show that our proposed adaptation-aware FSIG method outperforms SOTA FSIG methods [3, 4, 5] and produces high quality images with good diversity.

We further show 10-shot adaptation results for our proposed adaptation-aware FSIG method for additional setups. We show 10-shot adaptation results for FFHQ → MetFaces [8] (Figure S21), FFHQ → Sketches (Figure S22), FFHQ → Sunglasses (Figure S20), FFHQ → Amedeo Modigliani's Paintings (Figure S23), FFHQ → Otto Dix's Paintings (S24) and Cars → Wrecked Cars (Figure S26).

## F.3 100-shot adaptation

In addition to the analysis of increasing the number of shots for target adaptation in Figure 6 of main paper, here we additionally show the generated images with 100-shot training data, on Babies and AFHQ-Cat. The results are shown in Figure S27 where each column represents a fixed noise. Compared to baseline and SOTA methods, our generated images can still produce the best quality and diversity.

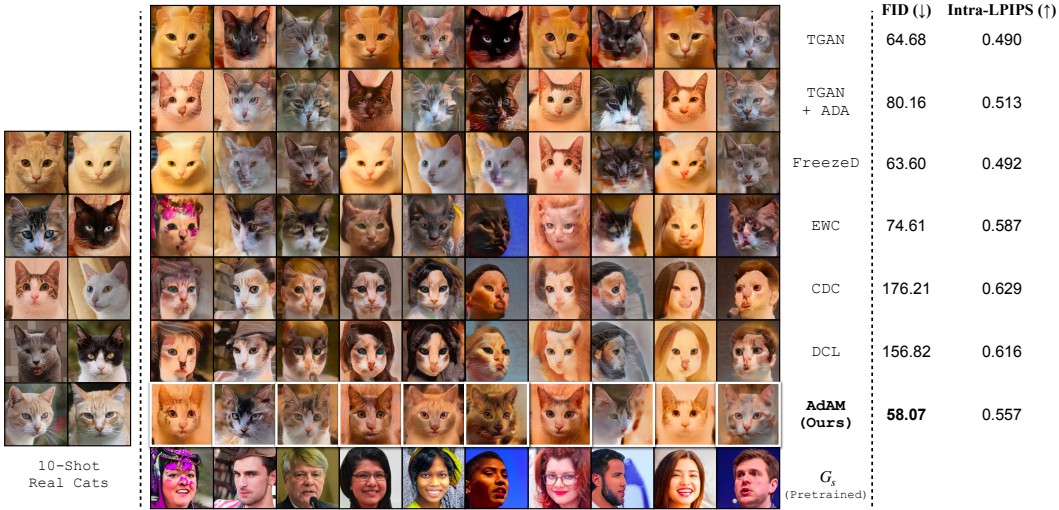

| | | FID (↓) | Intra-LPIPS (↑) |
|---|---|---|---|
| | TGAN | 64.68 | 0.490 |
| | TGAN + ADA | 80.16 | 0.513 |
| | FreezeD | 63.60 | 0.492 |
| | EWC | 74.61 | 0.587 |
| | CDC | 176.21 | 0.629 |
| | DCL | 156.82 | 0.616 |
| | **AdAM (Ours)** | **58.07** | 0.557 |
| 10-Shot Real Cats | $G_s$ (Pretrained) | | |

Figure S13: FFHQ → Babies results, including ADA [8].

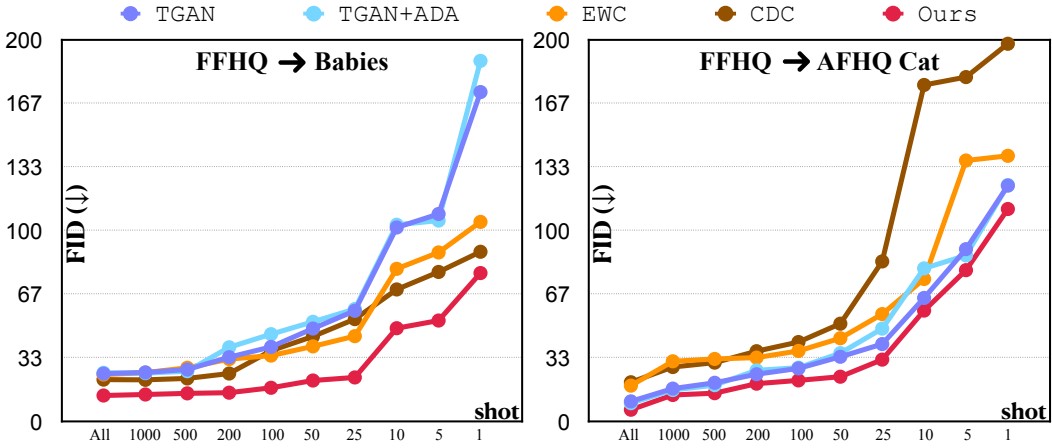

Figure S14: We add more data points based on Figure 6 in the main paper, and conduct experiments given extremely limited number of samples. We also include the entire dataset for adaptation.

### F.4 FID measurements with limited target domain samples

To characterize source → target domain proximity, we used FID and LPIPS measurements. FID involves distribution estimation using first-order (mean) and second-order (trace) moments, i.e.: $FID = mean_{component} + trace_{component}$ [30] Generally, 50K real and generated samples are used for FID calculation [2]. Given that our target domain datasets contain limited samples, i.e.: Cat [14], Dog [14], Wild [14] datasets contain ≈ 5K samples, we conduct extensive experiments to show that FID measurements with limited samples give reliable estimates, thereby reliably characterizing source → target domain proximity. Specifically, we decompose FID into mean and trace components and study the effect of target domain sample size to show that our proximity measurements using FID are reliable.

**Experiment Setup.** We use 3 large datasets namely FFHQ [18] (70K samples), LSUN-Bedroom [31] (70K samples) and LSUN-Cat [31] (70K samples). We use FFHQ (70K samples) as the source domain and study the effect of sample size on FID measure. Specifically, we decompose FID into mean and trace components in this study. We consider FFHQ (self-measurement), LSUN-Bedroom and LSUN-cat as target domains. We sample 13, 130, 1300, 2600, 5200, 13000, 52000 samples from

---

[2]Chong, Min Jin, and David Forsyth. "Effectively unbiased fid and inception score and where to find them." Proceedings of the IEEE/CVF conference on computer vision and pattern recognition. 2020.

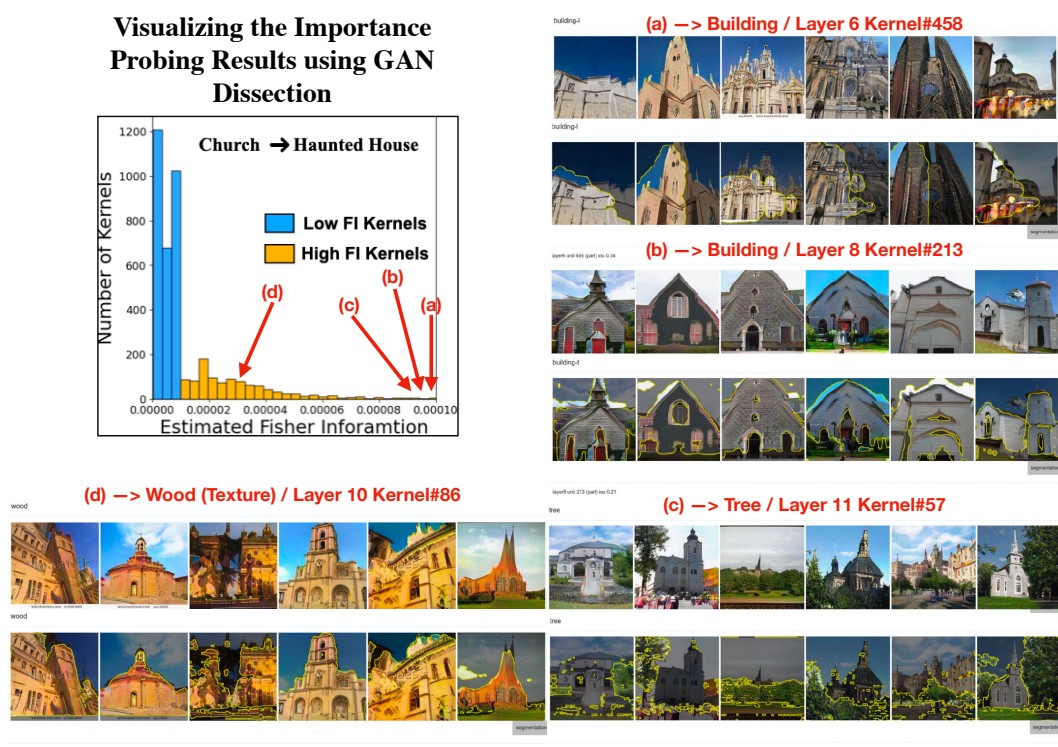

Figure S15: Visualizing high FI kernels using GAN Dissection [26] for Church → Haunted Houses 10-shot adaptation. In visualization of each high FI kernel, the first row shows different images generated by the source generator, and the second row highlights the concept encoded by the corresponding high FI kernel as determined by GAN Dissection. We observe that a notable amount of high FI kernels correspond to useful source domain concepts including building (a, b), tree (c) and wood (d) which are preserved when adapting to Haunted Houses target domain. We remark that these preserved concepts are useful to the target domain (Haunted House) for adaptation.

the target domain and measure the FID with FFHQ (70K samples), and compare it against the FID obtained by using the entire 70K samples from the target domain.

**Results / Analysis.** The results are shown in Table S7. As one can observe, with ≈ 2600 samples, we can reliably estimate FID as it becomes closer to the FID measured using the entire 70K target domain samples. Hence, we show that our source → target proximity measurements using FID are reliable.

## G   Discussion: How much can the proximity between source and target be relaxed?

In this section, we explore the proximity limitation between source and target domains in our experiment setups. First, we remark that the upper bound on proximity between the source domain S and the target domain T could be conditioning on (a) the number of available samples (shots) from the target domain, and (b) the method used for knowledge transfer.

(a) Proximity bound conditioning on the number of target domain samples. In this paper, we focus on few-shot setups, e.g. 10 shots. However, with more target domain samples available, proximity between S and T can be further relaxed, and the proximity bound would increase, i.e. for a given generative model on S, we could learn an adapted model for T which is more distant. Intuitively, increasing the number of target domain samples can provide more diverse knowledge for T, and as a result, there is less reliance on the knowledge of S that is generalizable for T (which would decrease as S and T are more apart). In the limiting cases when abundant target domain samples are available,

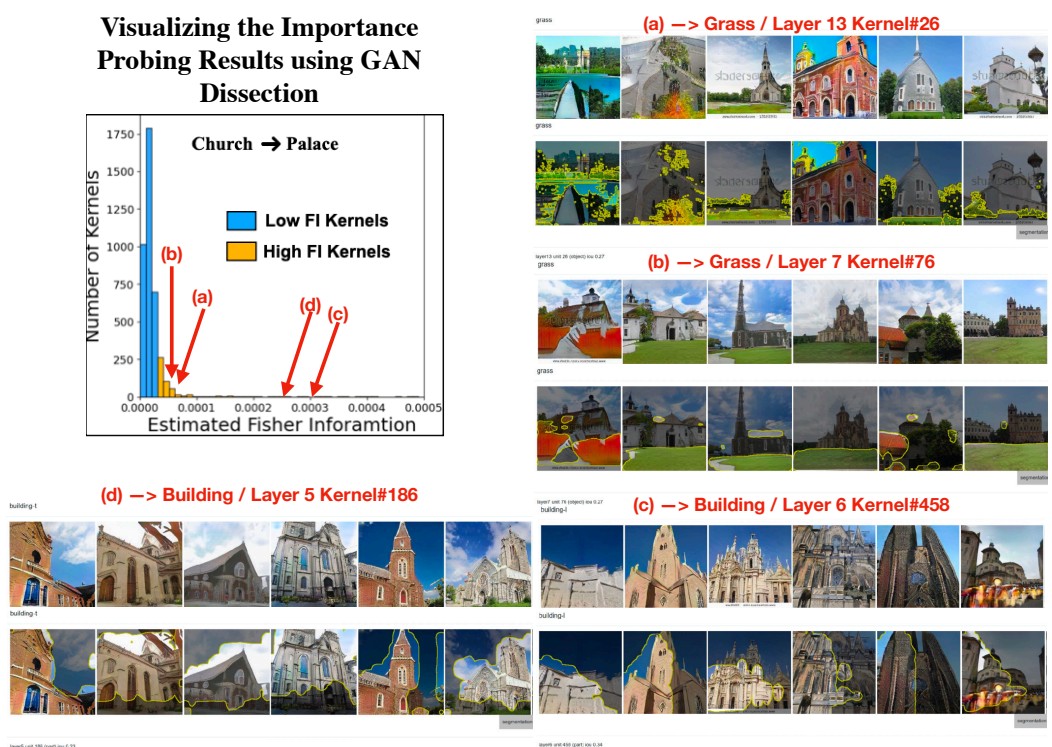

Figure S16: Visualizing high FI kernels using GAN Dissection [26] for Church → Palace 10-shot adaptation. In visualization of each high FI kernel, the first row shows different images generated by the source generator, and the second row highlights the concept encoded by the corresponding high FI kernel as determined by GAN Dissection. We observe that a notable amount of high FI kernels correspond to useful source domain concepts including grass (a, b) and building (c, d) which are preserved when adapting to Palace target domain. We remark that these preserved concepts are useful to the target domain (palace) for adaptation.

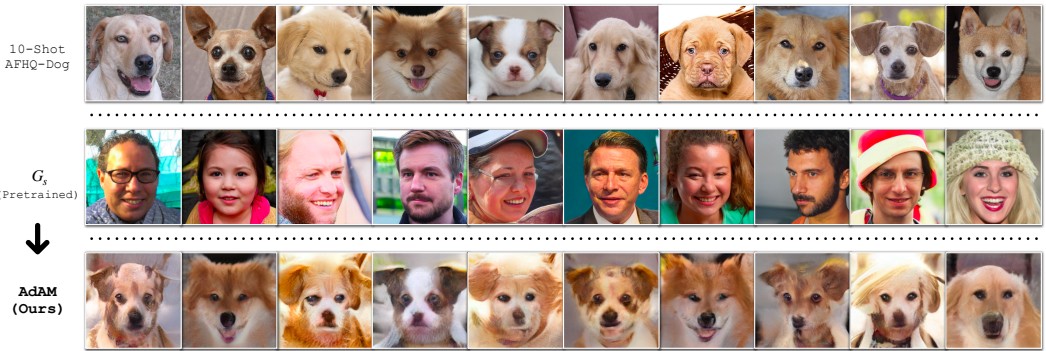

Figure S17: FFHQ → AFHQ-Dog (distant domain)

knowledge of S would not be critical, and proximity constraints between S and T may be totally relaxed (ignored).

(b) Proximity bound conditioning on the knowledge transfer method. Given a generative model pretrained on S and a certain number of available samples from T, the method used for knowledge transfer plays a critical role. If the method is superior in identifying suitable transferable knowledge from S to T, the proximity between S and T can be relaxed, and the proximity bound would increase. In our work, our first contribution is to reveal that existing SOTA approaches (which are based on target-agnostic ideas) are inadequate in identifying transferable knowledge from S to T. As a result,

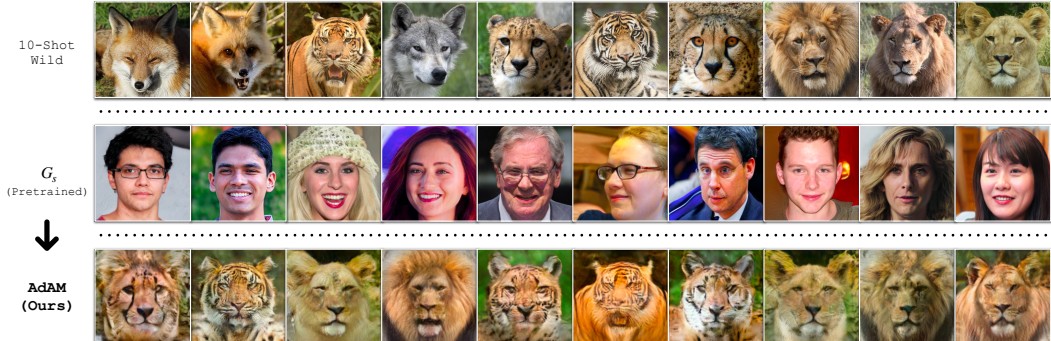

Figure S18: FFHQ → AFHQ-Wild (distant domain)

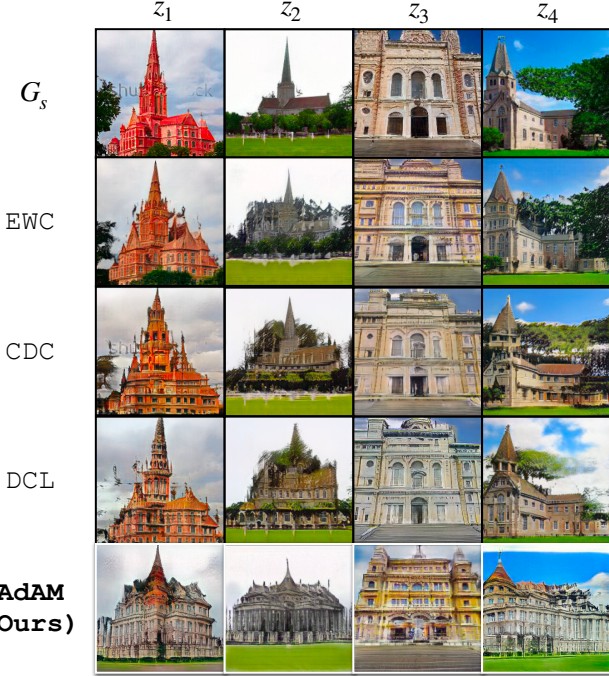

Figure S19: Church → Palace (distant domain)

when proximity between S and T is relaxed, the performance of the adapted models is miserably poor, as discussed in Sec.3, Sec.5, and Appendix. Therefore, our second contribution is to propose a target-aware approach that could identify more meaningful transferable knowledge from S to T, allowing relaxation of the proximity constraint.

In this section, we provide experimental results for the adaptation between two very distant domains: FFHQ→Cars using only 10-shots, aiming to answer two main questions: (1) Is there transferable knowledge from FFHQ to Cars for the FSIG task? (2) How does our proposed method compare with other methods in this setup? For this, in addition to transfer learning approaches discussed in the paper, we also add the results for training from scratch using only the same 10 Car samples. The quantitative results are in Table S8.

The results suggest that even though domain FFHQ and domain Cars are apart, there is still useful and transferable knowledge from FFHQ to Cars (e.g. low-level edges, shapes), leading to better performance (FID, Intra-LPIPS) in the adapted model using proposed method compared to the one which is trained from scratch. In addition, our proposed method can identify and transfer more meaningful knowledge compared to other baselines and SOTA methods, resulting in lower FID and higher diversity in generated images.

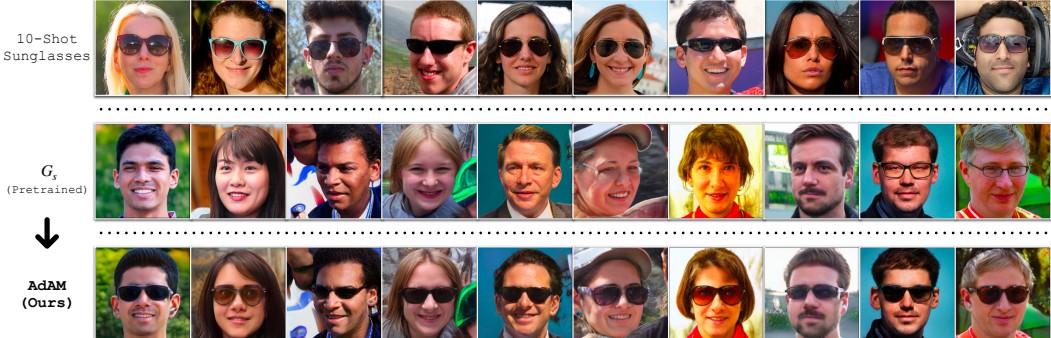

Figure S20: FFHQ → Sunglasses

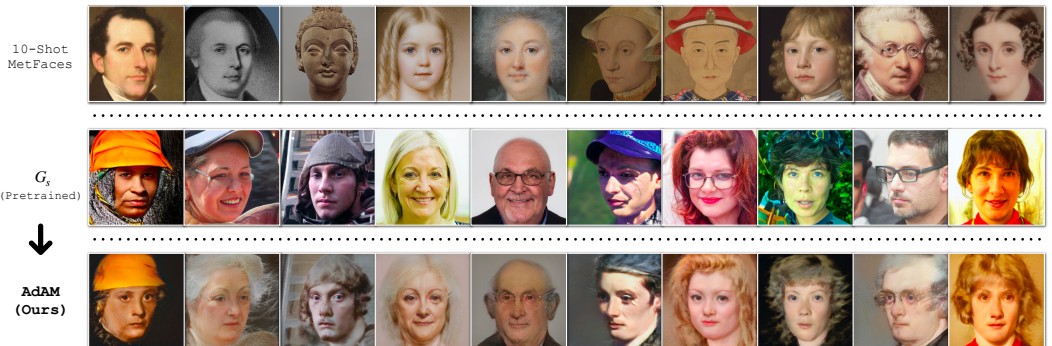

Figure S21: FFHQ → MetFaces

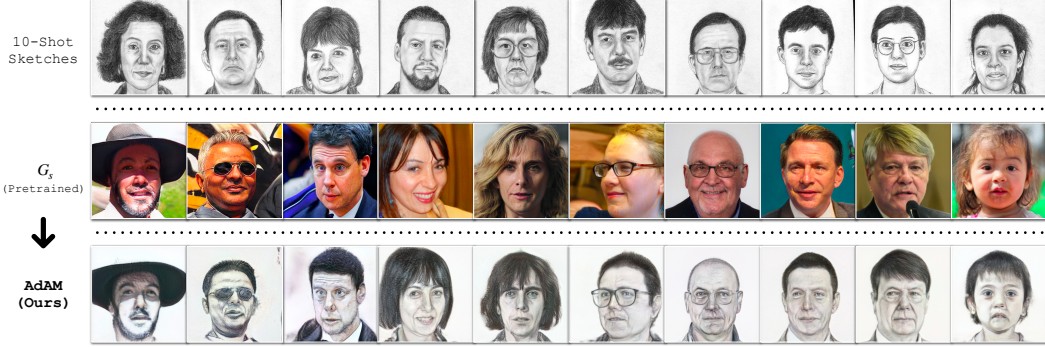

Figure S22: FFHQ → Sketches

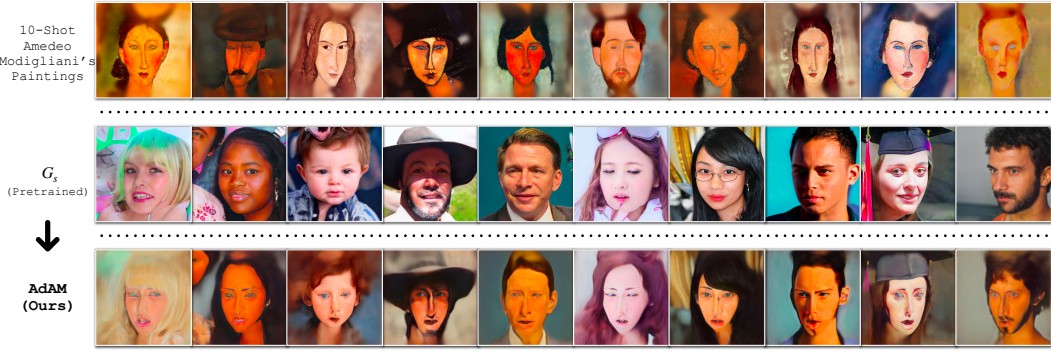

Figure S23: FFHQ → Amedeo Modigliani's Paintings

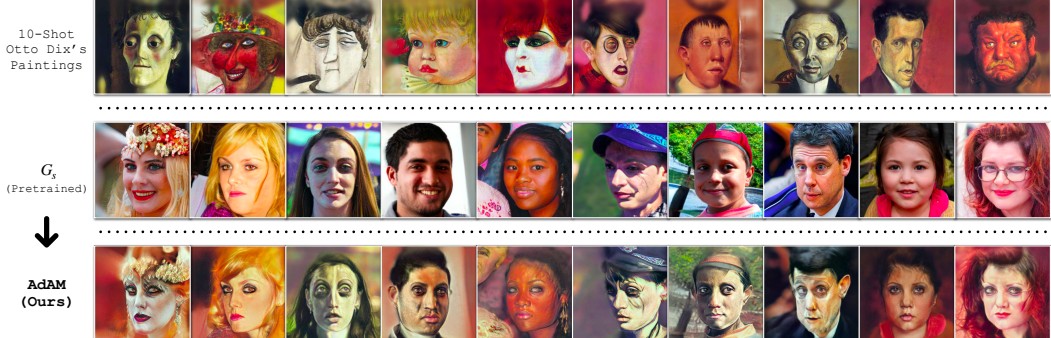

Figure S24: FFHQ → Otto Dix's Paintings

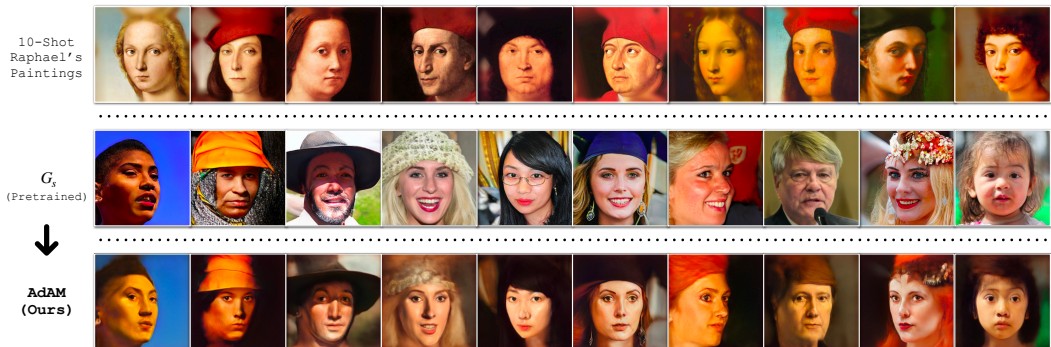

Figure S25: FFHQ → Raphael's Paintings

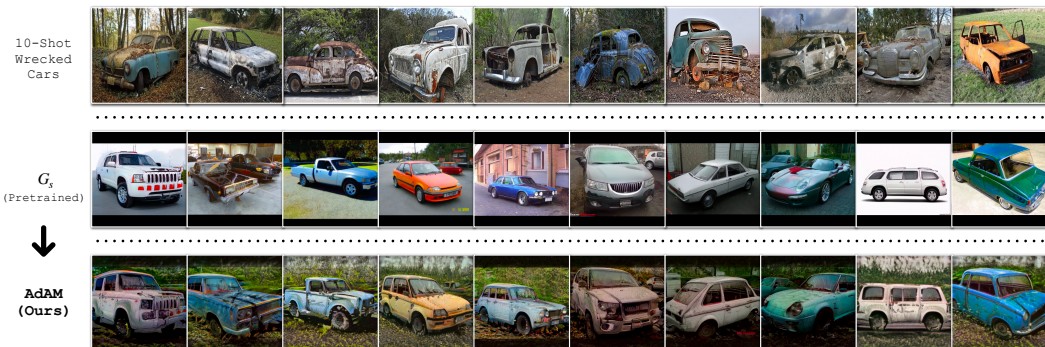

Figure S26: Cars → Wrecked Cars

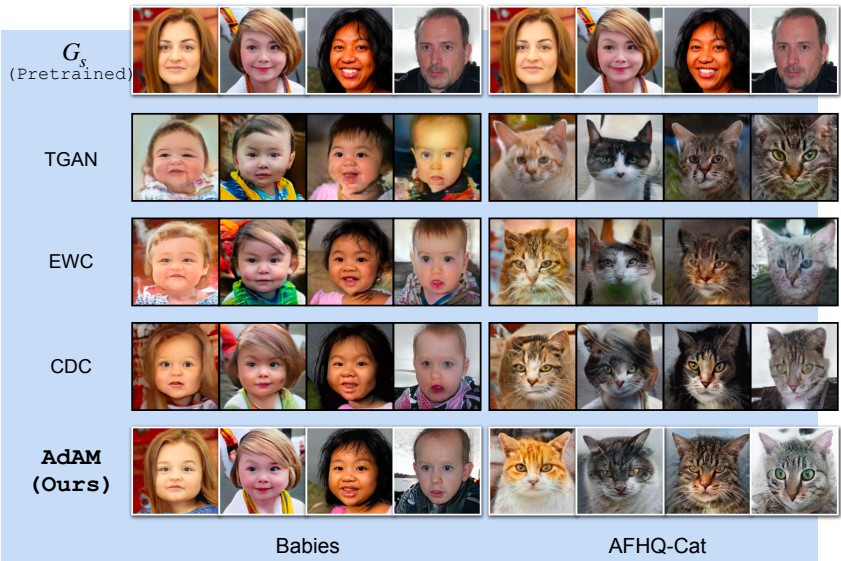

Figure S27: FFHQ → Babies (Left) and AFHQ-Cat (Right) with 100 samples for adaptation.

Table S7: *FID measurements with limited target domain samples give reliable estimates to characterize source → target domain proximity:* FFHQ (70K) is the source domain. We use FFHQ (self-measurement), LSUN-Bedroom and LSUN-Cat as target domains. We use different number of samples from target domain to measure FID. We also decompose FID into mean and trace components in this study. We sample 13, 130, 1300, 2600, 5200, 13000, 52000 images from the target domain and measure the FID with source domain (FFHQ / 70K), and compare it against the FID obtained by using the entire 70K samples from the target domain. Each experiment is repeated *100 times* and we report the results with standard deviation. We also report mean and trace components separately. As one can observe, with ≈ 2600 samples, we can reliably estimate FID as it becomes closer to the FID measured using the entire 70K target domain samples. Therefore, this study shows that our source → target proximity measurements using FID are reliable.

| FID | | 13 | 130 | 1300 | 2600 | 5200 | 13, 000 | 52, 000 | 70, 000 |
|---|---|---|---|---|---|---|---|---|---|
| FFHQ | FID | 196.3 ± 11.8 | 83.4 ± 2.2 | 15.3 ± 0.2 | **7 ± 0.1** | 3.3 ± 0 | 1.2 ± 0 | 0.1 ± 0 | 0 ± 0 |
| | mean | 12 ± 2.7 | 1.3 ± 0.3 | 0.1 ± 0 | **0.1 ± 0** | 0 ± 0 | 0 ± 0 | 0 ± 0 | 0 ± 0 |
| | trace | 184.3 ± 10.3 | 82.2 ± 2 | 15.2 ± 0.2 | **6.9 ± 0.1** | 3.3 ± 0 | 1.1 ± 0 | 0.1 ± 0 | 0 ± 0 |
| Bedroom | FID | 358.5 ± 9.3 | 301.9 ± 2.4 | 251 ± 1.2 | **243.6 ± 0.8** | 240.1 ± 0.5 | 238.2 ± 0.4 | 237.2 ± 0.2 | 237.2 ± 0.1 |
| | mean | 139.3 ± 8 | 131.8 ± 2.5 | 131.4 ± 0.9 | **131.1 ± 0.6** | 131.1 ± 0.4 | 131.1 ± 0.3 | 131.1 ± 0.1 | 131.1 ± 0.1 |
| | trace | 219.1 ± 9.9 | 170.1 ± 1.9 | 119.6 ± 0.6 | **112.5 ± 0.4** | 109.1 ± 0.3 | 107.1 ± 0.2 | 106.1 ± 0.1 | 106 ± 0.1 |
| Cat | FID | 370.2 ± 18.7 | 283.7 ± 4.4 | 209.7 ± 1.2 | **199.9 ± 0.8** | 195.3 ± 0.6 | 192.8 ± 0.4 | 191.4 ± 0.2 | 191.3 ± 0.1 |
| | mean | 105.7 ± 8.4 | 93 ± 2.2 | 91.7 ± 0.8 | **91.7 ± 0.5** | 91.6 ± 0.4 | 91.6 ± 0.2 | 91.6 ± 0.1 | 91.6 ± 0.1 |
| | trace | 264.5 ± 15.7 | 190.7 ± 3.6 | 118 ± 0.9 | **108.2 ± 0.6** | 103.7 ± 0.4 | 101.2 ± 0.3 | 99.9 ± 0.1 | 99.7 ± 0.1 |

# H    Additional information for Checklist

## H.1    Potential Societal Impact

Given very limited target domain samples (i.e.: 10-shot), our proposed method achieves SOTA results in FSIG with different source / target domain proximity. Though our work shows exciting results by pushing the limits of FSIG, we urge researchers, practitioners and developers to use our work with privacy, ethical and moral concerns. In what next, we bring an example to our discussion.

**Adapting the pretrained GAN to a particular person.** The idea of FSIG aims to adapt a pretrained GAN to a target domain with limited samples. It is reasonable that a user of FSIG can take few-shot images of a particular person and generate diverse images of the person, which leads to potential safety and privacy concerns. We conduct an experiment to adapt a pretrained StyleGAN2 generator to Obama dataset [32] in 1-shot, 5-shot and 10-shot setups, and the results are shown in S28.

Table S8: We conduct experiments for FFHQ → Cars adaptation and evaluate the performance in such a challenging setup. We show that our method can achieve similar diversity as ADA [8] and the overall performance (FID) is better than other baseline and SOTA methods.

| Domain | FFHQ → Cars | |
|---|---|---|
| | FID ($\downarrow$) | Intra-LPIPS ($\uparrow$) |
| Train from Scratch | 201.34 | 0.300 |
| ADA [8] | 171.98 | 0.438 |
| EWC [3] | 276.19 | **0.620** |
| CDC [4] | 109.53 | 0.484 |
| DCL [5] | 125.96 | 0.464 |
| AdAM (Ours) | **80.55** | 0.425 |

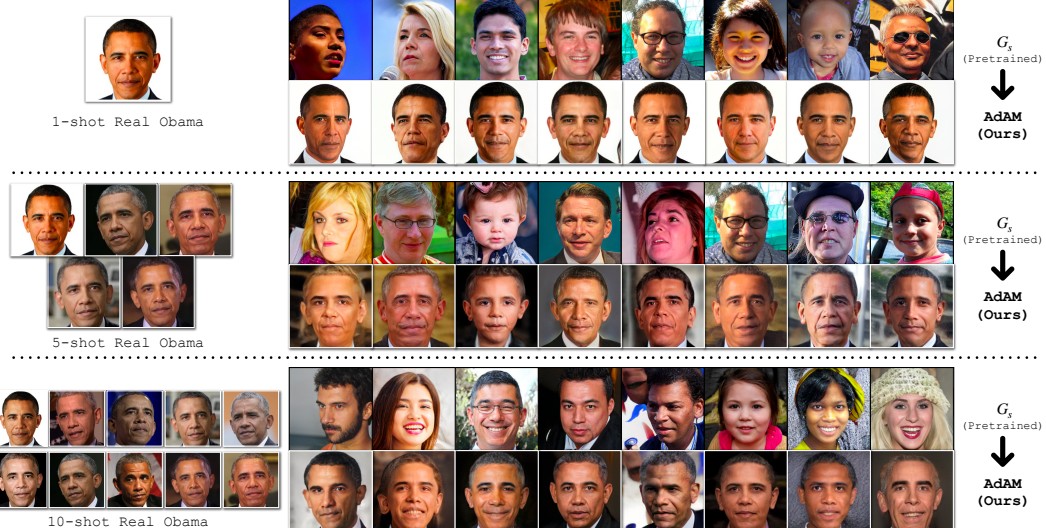

Figure S28: FFHQ → Obama Dataset

Potentially, the method can be adapted to generate images of the same person by applying a more restrictive selection of the source model's knowledge. However, this would degrade the diversity of the outputs and may not be suitable for general FSIG which our paper focuses on. We will explore such interesting application as our future work and verify with state-of-the-art face recognition systems to understand any potential threats.

## H.2 Amount of Compute

The amount of compute in this project is reported in Table S9. We follow NeurIPS guidelines to include the amount of compute for different experiments along with $CO_2$ emission.

Table S9: Amount of compute in this project. The GPU hours include computations for initial explorations / experiments to produce the reported values. $CO_2$ emission values are computed using https://mlco2.github.io/impact/

| Experiment | Hardware | GPU hours | Carbon emitted in kg |
|---|---|---|---|
| Main paper : Table 2 (Repeated 3 times) | Tesla V100-PCIE (32 GB) | 306 | 52.33 |
| Main paper : Figure 5 / Figure 6 | Tesla V100-PCIE (32 GB) | 136 | 23.26 |
| Main paper : Figure 2 | Tesla V100-PCIE (32 GB) | 6 | 1.03 |
| Supplementary : Extended Experiments | Tesla V100-PCIE (32 GB) | 68 | 11.63 |
| Supplementary : Ablation Study | Tesla V100-PCIE (32 GB) | 14 | 4.1 |
| Additional Compute for Hyper-parameter tuning | Tesla V100-PCIE (32 GB) | 24 | 2.16 |
| **Total** | | **554** | **94.73** |