# OpenReview forum: "Few-shot Image Generation via Adaptation-Aware Kernel Modulation"
_NeurIPS.cc/2022/Conference — NeurIPS 2022 Accept_

### Official Review · Reviewer_gbAJ · 2022-07-10

**Rating:** 6
**Confidence:** 3
**Soundness:** 3 good
**Presentation:** 3 good
**Contribution:** 3 good

**Summary:**

This paper proposes a method to do few-shot (e.g. 10 images) image generation, by transferring a pretrained GAN from a source domain (e.g. FFHQ) to a target domain (e.g. dogs/cats). The method is simple and the empirical performance is good.

**Questions:**

1.  I see connections between Fisher information (FI) and class saliency [a] or other similar attribution methods. But these attribution methods use first-order derivatives only, so they are much cheaper. Have you verified that FI is indeed advantageous over first-order attribution methods? In other words, Is using second-order derivatives indeed necessary?
2. In Fig. 2, the Sunglasses dataset is from [32]. But in remaining text, it's from [8]. However, I couldn't find the Sunglasses dataset from either of these two papers.
3. The references [32] and [33] are duplicate.
4. Why the first face (the young lady) of G_s in row 7 is the same as the 9th face of G_s in the last row?
5. Suggest the authors to use superscripts $\boldsymbol{m}_1^i$ instead of $\boldsymbol{m}_1(i)$ in Eq.(3) for better clarity.

[a] Deep Inside Convolutional Networks: Visualising Image Classification Models and Saliency Maps, 2014.

**Limitations:**

1. It is natural that a user of FSIG can take few-shot images of a particular person and generate diverse images of them. Have the authors tried this setting? How well does it work? If the performance is good, it could be a security threat to face recognition systems.

**Strengths And Weaknesses:**

Strengths:
1. Few-shot (e.g. 10 images) image generation is practically useful.
2. The empirical performance is good.

Weaknesses:
1. In eq.(3), the size of the fisher information matrix in (3) is square of the number of parameters, i.e., $(c_{in}\cdot k\cdot k)^2$. This could be huge. However, the authors claim this is very lightweight with minimal overhead. Please justify.
EDIT: I've read the appendix and found the derivations. However, $|\boldsymbol{m}_2|$ seems to mean the number of parameters in $\boldsymbol{m}_2$ (Eq.3 appendix)? But in the text before Eq.3 (main text), $|\boldsymbol{m}_2|=k$, the number of conv channels.
2. The chosen target-domain datasets, babies, metfaces, cat, dog, wild are actually quite similar to the source domain (human or animal faces). According to [33], the domain gaps between human and animal faces is not so big as they look.
3. In Fig. 4 and 6, the authors should include TGAN+ADA as an important baseline, instead of focusing on the much weaker TGAN. According to [33], when there are 1k target-domain images, the FID is quite low (around 25, Fig. 9d [33]), when adapting from FFHQ to Cat. So I expect that at the left side (200 shots) of Fig 6, TGAN+ADA should have competitive performance.
4. Table 2, Fig 5 and Fig 6 are crammed in Page 9.

---

> ### Author Response · Authors · 2022-08-02
> **[Response for Reviewer gbAJ] Part 3/3**
>
> > Suggest the authors to use superscripts $m_1^i$ instead of $m_1(i)$ in Eq.(3) for better clarity.
>
> Thanks for the suggestion. We will update the paper accordingly.
>
> $ $
>
> > **Limitation**: It is natural that a user of FSIG can take few-shot images of a particular person and generate diverse images of them. Have the authors tried this setting? How well does it work? If the performance is good, it could be a security threat to face recognition systems.
>
> The Reviewer's suggestion is very interesting. We conduct some experiments to understand this application. Results are shown in https://drive.google.com/drive/folders/14vTM_2vXF8YNRe50zyoy4kpD6xqnieBX?usp=sharing.
>
> As shown in the results, indeed we can adapt the model using few-shot images of a particular person (here we use the Obama dataset), and generate diverse images. Furthermore, the generated diverse images capture important facial characteristics of the particular person. As shown in our results, important facial characteristics of Obama (face shapes, gender, skin tone) are successfully learned by the adapted models.  It can also be observed that diverse source model’s knowledge meaningful for the target domain is preserved. Overall, the outputs of the adapted model resemble the particular person and have sufficient diversity such that they do not overfit the given training samples.
>
> Potentially, the method can be adapted to generate images of the same person by applying a more restrictive selection of the source model’s knowledge. However, this would degrade the diversity of the outputs and may not be suitable for general FSIG which our paper focuses on. We will look into this interesting application in our future work and verify with state-of-the-art face recognition systems to understand the potential threat. Note that, we have mentioned privacy concerns as a potential societal impact of our work, see Sec F1.
>
> We thank the Reviewer’s consideration on this and hope it has been clarified.

---

> ### Author Response · Authors · 2022-08-02
> **[Response for Reviewer gbAJ] Part 2/3**
>
> > Table 2, Fig 5, and Fig 6 are crammed on Page 9.
>
> We thank the Reviewer’s comment. Yes, we will fix the presentation issues of the figures and table.
>
> $ $
>
> > I see connections between Fisher information (FI) and class saliency [a] or other similar attribution methods. But these attribution methods use first-order derivatives only, so they are much cheaper. Have you verified that FI is indeed advantageous over first-order attribution methods? In other words, Is using second-order derivatives indeed necessary?
>
> Thank you for your comment. First, we remark that Fisher Information (FI) is not a fundamental part of our proposed target-aware adaptation approach, however, it is a theoretically grounded metric that we have selected to measure the importance of the kernels for the target task. The reason for our work to select FI as the metric to be used in importance probing is as follows: As explained in [44], FI provides a measure of the information a particular parameter contains about data distribution, i.e., parameters with higher FI values have higher importance for a given task. Therefore, FI is a principled method to measure the importance of the kernels for the adaptation task in our proposed approach.
>
> Importantly, we remark that as mentioned in the supplementary (Sec A.2, Page 3), following [44], we use the **square of the first-order derivatives** (instead of the second-order derivatives) to approximate the FI. Therefore, the complexity of our approximated FI is similar to other methods based on the first-order derivatives such as Class Saliency (CS) [A] mentioned by the Reviewer. We further remark that, as discussed in the supplementary (Sec A.1, Table S1), our proposed importance probing is very lightweight, requiring only ~8 minutes, thanks to our low-rank kernel modulation which reduces trainable parameters significantly.
>
> We agree with the Reviewer that conceptually CS could have a connection with FI as they both use the knowledge encoded in the gradients. As explained, we believe FI is a principled method for importance measurement. Based on the Reviewer’s comment, we perform an experiment to replace FI with CS in importance probing and compare it with our original approach. Note that, in [A], CS is computed w.r.t. input image pixels. To make CS suitable for our problem, we modify it and compute CS w.r.t. modulation parameters. Similar to our approach in the main paper, we average the importance of all parameters within a kernel to calculate the importance of that kernel. Then we use these values during our importance probing to determine the important kernels for adapting from source to target domain (as Sec. 4 in our main paper). The following results are obtained with our proposed method when using FI and CS during importance probing:
>
> **FFHQ$\rightarrow$Cat:**
>
> | Method        | FID($\downarrow$) | Intra-LPIPS($\uparrow$) |
> | ------------- |:-------------:|:-------------:|
> | Class Saliency [A] | 61.68 | 0.556  |
> | **Ours**: Fisher Information approximated by first-order derivatives | **58.07** | **0.557**  |
>
> **FFHQ$\rightarrow$Babies:**
>
> | Method        | FID($\downarrow$) | Intra-LPIPS($\uparrow$) |
> | ------------- |:-------------:|:-------------:|
> | Class Saliency [A] | 52.46 | 0.582  |
> | **Ours**: Fisher Information approximated by first-order derivatives | **48.83** | **0.590**  |
>
>
> Our results suggest that importance probing using FI (approximated by first-order derivatives) can perform better in the selection of important kernels, leading to better performance (FID, intra-LPIPS) in the adapted models as shown in the table. Please also note that the running times of the two methods during importance probing are very close,  e.g., for FFHQ$\rightarrow$Cat: ours (8 min 43s) vs CS (9 mins 58s). The corresponding model checkpoints in the above table can be found here: https://drive.google.com/drive/folders/1X6W7KD91wQSSqtLdrdfD_Ui4vcIaEHqH?usp=sharing).
>
> [A] Deep Inside Convolutional Networks: Visualising Image Classification Models and Saliency Maps, 2014.
>
>
> $ $
>
> > In Fig. 2, the Sunglasses dataset is from [32]. But in remaining text, it's from [8]. However, I couldn't find the Sunglasses dataset from either of these two papers.
>
> Thanks for pointing out this. The Sunglasses dataset is extracted from Ojha et al. CDC paper [4]. We will update the submission to fix these typos.
>
> $ $
>
> > The references [32] and [33] are duplicate.
>
> Thanks for pointing out this. Yes, we will fix the typos.
>
> $ $
>
> > Why the first face (the young lady) of G_s in row 7 is the same as the 9th face of G_s in the last row?
>
> Thanks for your comment. In these two cases, the same noise input codes are used as input to the generator, therefore, the generated images (the young lady in this case) *before adaptation to any target domain* are the same. Note that, Row 7 and Row 14 show the outputs of the *pretrained* generators, and the same pretrained generators are used in these cases.

---

> ### Author Response · Authors · 2022-08-02
> **[Response for Reviewer gbAJ] Part 1/3**
>
> We appreciate the valuable comments of the Reviewer. The Reviewer’s opinion about the importance of the topic, and the empirical performance of our method is valuable for us. We have provided the details/experimental results for each comment raised by the Reviewer, and hope these details address the concerns of the Reviewer.
>
> $ $
>
> > I've read the appendix and found the derivations. However, $|m_2|$ seems to mean the number of parameters in $m_2$ (Eq.3 appendix)? But in the text before Eq.3 (main text), $|m_2|=k$, the number of conv channels.
>
> Thank you for the comment. Yes, indeed we use only diagonal entries (square of the gradients) to estimate FI instead of the full matrix to keep the importance-probing lightweight.
>
> Thank you for pointing out the typo. Yes, $|m_2|$ should be $c_{in} \times k \times k$. We will fix that.
>
> $ $
>
> > The chosen target-domain datasets, babies, metfaces, cat, dog, wild are actually quite similar to the source domain (human or animal faces). According to [33], the domain gaps between human and animal faces is not so big as they look.
>
> Thank you for the comment. To the best of our knowledge, in the ADA [33] there is no analysis regarding the domain distances. There are only experimental results in Figure 11 (a) of [33] which suggest: “*When 5153 images are available (i.e., the entire dataset) from AFHQ-Cat, transfer learning from FFHQ to AFHQ-Cat (with ADA) achieves better results than training from scratch on AFHQ-Cat (with ADA)*.” This indicates that there is some transferable knowledge from FFHQ to AFHQ-Cat, which our work leverages to learn generative models under few-shot setups. However, we believe [33] gives us no information about the domain gap between the source domain and different target domains (e.g., the domain gap between “FFHQ$\rightarrow$Cat” vs “FFHQ$\rightarrow$Babies”).
>
> Note that in Figure 2 of our main paper, we have analyzed the distance between different domains both qualitatively (using Inception-v3 and LPIPS features) and quantitatively (using FID and LPIPS distances) to show that current FSIG methods focus on target domains that are proximal to the source domain FFHQ (e.g., FFHQ$\rightarrow$Babies). Our analysis shows both quantitatively and qualitatively that as an example, the domain gap between FFHQ and AFHQ-Cat (FID=227, and larger visual distance) is larger than the domain gap between FFHQ and babies (FID=147, and smaller visual distance).
>
> Please also note that when transferring the knowledge from a source domain to a target domain (e.g., FFHQ$\rightarrow$Cat) using a specific transfer learning approach, the hardness of overcoming the domain gap depends on the number of samples. The more samples available from the target domain, generally it is easier to obtain domain-specific knowledge and prevent distorting general knowledge of the pre-trained model. Therefore, both results in ADA [33] and our paper confirms transferability, however, our problem setup (10 shots adaptation) is much more challenging compared to ADA (5153 shots adaptation).
>
> $ $
>
> > In Fig. 4 and 6, the authors should include TGAN+ADA as an important baseline, instead of focusing on the much weaker TGAN. According to [33], when there are 1k target-domain images, the FID is quite low (around 25, Fig. 9d [33]), when adapting from FFHQ to Cat. So I expect that at the left side (200 shots) of Fig 6, TGAN+ADA should have competitive performance.
>
> We thank the Reviewer’s comment and we agree that ADA is an important baseline in training GANs with limited data. We remark that we have included the performance of TGAN+ADA in Table 2 of the main paper. Additionally, we further note that TGAN+ADA performs rather poorly when there are only a few target samples (e.g., 10-shot), and sometimes it may lead to leakage of the augmentation to the generated images (this is also observed in Ojha et al. [4], Figure 4 row 2).
>
> Nevertheless, we follow the Reviewer’s suggestion and conduct detailed experiments:
> * **Figure 4:** We show generated images of TGAN+ADA, and compare them with other methods qualitatively and quantitatively (FID and intra-LPIPS). Results can be found here: https://drive.google.com/drive/folders/1S3EWTMKjKvpsZ2RU-2ZreVw0Sd0crgsM?usp=sharing.
> * **Figure 6**: We evaluate TGAN+ADA in our experiments with different amounts of target data and compare it with other methods. We show that when the target domain contains sufficient training samples (e.g., 1K or the entire dataset), TGAN+ADA is indeed a strong baseline method.  Results can be found here: https://drive.google.com/drive/folders/1LHfRYZ6oCOhKL9KN_fAQEx7hhtQIn9c-?usp=sharing. We humbly point out that the FID score in Fig. 9d [33] is from FFHQ$\rightarrow$**LSUN-Cat**, but not FFHQ$\rightarrow$**AFHQ-Cat** in our experiments.
>
> In all experiments, we show that our methods can consistently outperform TGAN+ADA qualitatively and quantitatively.

---

> > ### Comment · Reviewer_gbAJ · 2022-08-08
> > **Thanks for the response**
> >
> > Generally I'm happy with the extra results and explanations. Would raise my rating to weak accept.

---

### Official Review · Reviewer_4Rin · 2022-07-10

**Rating:** 7
**Confidence:** 3
**Soundness:** 4 excellent
**Presentation:** 3 good
**Contribution:** 3 good

**Summary:**

This paper tackle the problem of few-shot image generation, which aims to learn to generate image given a limited number of samples from a domain. The author first conducts a probing analysis of existing methods and finds out the reasons for degenerated performance of these methods under the assumption of relaxed proximity between source and target domains. Secondly, the author propose adaptation-aware kernel modulation to address general FSIG of different source-target domain proximity. This results in good performance, in terms of generated image quality in multiple few-shot settings.

**Questions:**

1. More analysis of the proposed importance probing is recommended. The author proposes important probing to identify the importance of each kernel in the GAN model. To further verify the generalizability of this probing method, I would suggest the authors try the importance probing on different types of pre-trained GAN models.

2. Is it possible to visualize the importance probing results in Figure 5? In this way, we are able to reveal what kind of information is preserved (w/ high IF) in the modulation, and this can serve as support for the effectiveness of this probing analysis.

3. More discussion on (potential) failure cases is recommended in the main paper or appendix.

Minor comment:

1. The layout of the main paper should be reorganized in the future version. The margins between some figures and tables (e.g., Figure 5 and 6) are too small, making the captions a bit hard to read.

**Limitations:**

Yes, the author discusses the potential societal impact in Section F1 of the appendix.

**Strengths And Weaknesses:**

Strengths:

1. The paper is well-written and easy to follow.

2. There are good performance gains. The proposed approach is (to the best of my knowledge) novel, and also theoretically interesting.

3. The model is evaluated on multiple few-shot settings and compared to multiple baselines, which shows its outperformance. The proposed method obtains SOTA on few-shot image generation performance, with a good ablation study.

4. Code is provided, and reproducibility is considered.

Weaknesses:

(Please refer to "Questions".)

---

> ### Author Response · Authors · 2022-08-02
> **[Response for Reviewer 4Rin] Part 2/2**
>
> $ $
>
> > More analysis of the proposed importance probing is recommended. The author proposes important probing to identify the importance of each kernel in the GAN model. To further verify the generalizability of this probing method, I would suggest the authors try the importance probing on different types of pre-trained GAN models.
>
> We thank the Reviewer’s insightful comment regarding generalizability. Following recent works [4][5], we choose StyleGAN-V2 as the GAN architecture in our main paper. Our proposed idea is a general framework and can be applied to other pretrained GAN models.
>
> Following the Reviewer’s suggestion, we include the experiment results using **ProGAN** [A] as another pre-trained GAN model. Similar to the experimental setup in our submission, we use the ProGAN pretrained on FFHQ and LSUN Church, and adapt it to different target domains with 10 shots, as listed below:
> * FFHQ$\rightarrow$Babies (source and target domains are **close**)
> * FFHQ$\rightarrow$AFHQ-Cat (source and target domains are **distant**)
> * Church$\rightarrow$Haunted House (source and target domains are **close**)
> * Church$\rightarrow$Palace (source and target domains are **distant**)
>
> We evaluate our proposed method and compare it with other methods qualitatively and quantitatively. The experimental results can be found in: https://drive.google.com/drive/folders/1hPI6cxkfW6OGznrshkP8cnFgAP0NtsWA?usp=sharing.
>
> We further provide code and checkpoints for **reproducible research**:
> * The estimated Fisher Information Dictionary of ProGAN: https://drive.google.com/drive/folders/1hHvaIeDOKfwUmW8iUtJgDp5MxLzeYY8a?usp=sharing
> * ProGAN checkpoints corresponding to the above visualization:https://drive.google.com/drive/folders/1Y28rwmT5i-mHnUPRh3ChRwZlvW6pjVsO?usp=sharing
> * Code implementation: https://drive.google.com/drive/folders/1aVfKnUIRKmFHODGeF4vvcNeMZC7H5E8A?usp=sharing
>
> In summary, these experimental results verify that similar to Figure 4 and Figure S9 in our submission, **our proposed method can also consistently outperform other baseline and SOTA methods with another pre-trained GAN model (ProGAN)**, demonstrating the **_effectiveness and generalizability_** of our method.
>
> [A] Karra, et al. "Progressive Growing of GANs for Improved Quality, Stability, and Variation."  In ICLR 2018.
>
> $ $
>
>
> > More discussion on (potential) failure cases is recommended in the main paper or appendix.
>
> We thank the Reviewer’s comment. We will include more discussion on failure cases. In particular, we perform experiments on the very challenging setups of FFHQ$\rightarrow$Cars (pre-trained model: StyleGANv2). We evaluate our method and other existing state-of-the-art approaches. Some generated samples can be found in this anonymous link:
> https://drive.google.com/drive/folders/1a_ENA1wcqBl5vKrD_MfqVODktmIJKbt6?usp=sharing. The visual results show that there is room for improvement under this setup when the distance between source/target domains is further increased. Some failure cases using our method can be observed (and also for all other methods).
>
> We will include this experiment and discuss the failure cases in the paper.
>
> $ $
>
> > The layout of the main paper should be reorganized in the future version. The margins between some figures and tables (e.g., Figure 5 and 6) are too small, making the captions a bit hard to read.
>
> Thank you for your comment, We will reorganize the layout of the paper to fix these problems in the next version.

---

> ### Author Response · Authors · 2022-08-02
> **[Response for Reviewer 4Rin] Part 1/2**
>
> We thank the Reviewer for the positive feedback on our work. We have provided further details including experiments with another GAN structure, the visualization of the importance-probing, and the discussion on the failure cases. We hope provided results address the concerns of the Reviewer.
>
> $ $
>
> > Is it possible to visualize the importance probing results in Figure 5? In this way, we are able to reveal what kind of information is preserved (w/ high IF) in the modulation, and this can serve as support for the effectiveness of this probing analysis.
>
> We appreciate Reviewer’s insightful comment. We remark that visualizing concepts/semantics learned by generative models is not straightforward, i.e.: *discovering what form of visual information is encoded/generated by a specific high FI kernel is a complex problem*. To our best knowledge, methods of visualizing generative models/GANs are still rather restrictive in terms of concepts that can be visualized. Nevertheless, we respect Reviewer’s suggestion and leverage on **GAN Dissection** [A], an established visualization method to visualize the high FI internal representations (corresponding to results in Figure 5). The experiment details are as follows.
>
> **Experiment setup**: We use Church as source domain as the [official GAN Dissection method](https://github.com/CSAILVision/gandissect) is more suitable for scene-based image generation models (due to the limitation of the semantic segmentation pipeline in GAN Dissection [A]). Following our submission, we use two target domains: Haunted Houses (proximal domain) and Palace (distant domain). We remark that proximity analysis and qualitative results for Church$\rightarrow$Haunted Houses, and Church$\rightarrow$Palace are included in the original submission (Supplementary Figures S2, S3, S5). Following official GAN Dissection implementation [A], we use the ProGAN [C] model. We use 256 x 256 resolution for adaptation.
>
> The following results are obtained:
> * **Visualizing high FI kernels for Church$\rightarrow$Haunted Houses adaptation:** The results for FI estimation for kernels and several distinct semantic concepts learned by high FI kernels are shown here: https://drive.google.com/drive/folders/19UdNYTG2SMqDPx3UjQtnfcRAOk0mFXHu?usp=sharing. In the Figure, we visualize four examples of high FI kernels: (a), (b), (c), (d) corresponding to concepts building, building, tree, and wood respectively. In the visualization of each high FI kernel, first row shows different images generated by source generator, and second row highlights concept encoded by the corresponding high FI kernel as determined by GAN Dissection.  Using GAN Dissection, we observe that a notable amount of high FI kernels correspond to useful source domain concepts including *building, tree and wood (texture)* which are preserved when adapting to Haunted Houses target domain. We remark that these preserved concepts are useful to the target domain for adaptation. These results add merit to our findings and will be included in the next version.
>
> * **Visualizing high FI kernels for Church$\rightarrow$Palace adaptation:** The results for FI estimation for kernels and several distinct semantic concepts learned by high FI kernels are shown here: https://drive.google.com/drive/folders/19UdNYTG2SMqDPx3UjQtnfcRAOk0mFXHu?usp=sharing. In the Figure, we visualize four examples of high FI kernels: (a), (b), (c), (d) corresponding to concepts grass, grass, building, and building respectively. In visualization of each high FI kernel, first row shows different images generated by source generator, and second row highlights concept encoded by the corresponding high FI kernel. Similarly, we observe that a notable amount of high FI kernels correspond to useful source domain concepts including *grass and building* which are preserved when adapting to the Palace domain. We remark that these preserved concepts are useful to the target domain for adaptation. These results add merit to our findings and will be included in next version.
>
> **Limitations of GAN Dissection/Future Work:** Although GAN Dissection can uncover useful semantic concepts preserved by high FI kernels, the GAN Dissection method [A] is limited by dataset used for semantic segmentation. Hence this method is not able to uncover concepts that are not present in semantic segmentation datasets (They use Broaden Dataset [B]). Therefore, using GAN dissection we are currently unable to discover and visualize more fine-grained concepts preserved by our high FI kernels. We hope to further address this problem in future work. We thank the reviewer for the valuable comment.
>
> [A] Bau, David, et al. "GAN Dissection: Visualizing and Understanding Generative Adversarial Networks." ICLR 2019.
>
> [B] Bau, David, et al. "Network dissection: Quantifying interpretability of deep visual representations." CVPR 2017.
>
> [C] Karras, Tero, et al. "Progressive Growing of GANs for Improved Quality, Stability, and Variation." ICLR 2018.

---

> ### Author Response · Authors · 2022-08-08
> **We thank the valuable time from Reviewer 4Rin**
>
> Dear Reviewer **4Rin** :
>
> We sincerely thank you for your valuable time.
>
> In our previous responses, we have tried our best to address all questions by Reviewer **4Rin**. We would like to ask if our responses are adequate. If not, we would be happy to provide more information.
>
> In particular, we note that our additional requested results further strengthen the main findings of our paper, and we have tried to answer Reviewer **4Rin**’s questions comprehensively:
>
> - We apply our proposed FSIG method to **another GAN architecture: ProGAN**, and compare with different baselines and SOTA FSIG methods. We show that our proposed Adaptation-aware FSIG method  performs better than existing FSIG methods consistently;
>
> - We **visualize the important kernels** identified in the probing stage via “GAN Dissection”. We show that important kernels indeed encode meaningful features useful for the target domain.
>
> **For all experiments in the submission and rebuttal, we provide the Pytorch code/checkpoints for reproducible research.**
>
> Additionally, we thank Reviewer **4Rin** for the very positive score especially the **"excellent"** rating for "Soundness", and **"good"** for “Presentation” and “Contribution”, with an **overall Rating “7 Accept”:**
>
> - **Soundness:** 4 excellent
> - **Presentation:** 3 good
> - **Contribution:** 3 good
> - **Rating:** 7 Accept
>
> Given the positive assessment and rating, with our additional response and further results, **we would like to humbly request Reviewer **4Rin** to consider increasing the overall rating if there is no further question.**
>
> We thank you for your valuable time and consideration.

---

> ### Comment · Reviewer_4Rin · 2022-08-08
> **Thanks for the response!**
>
> Thank the authors for the detailed clarification and comments. Overall, the paper is interesting, well written, and constitutes a good scientific contribution in my opinion. I gave a higher score compared to the other reviewers and still stand by my initial assessment.

---

### Official Review · Reviewer_ekht · 2022-07-11

**Rating:** 6
**Confidence:** 5
**Soundness:** 4 excellent
**Presentation:** 4 excellent
**Contribution:** 3 good

**Summary:**

This paper focuses on the few-shot image generation, especially when there might be a huge gap between the source and target domains. By setting up a valiation pipelline, authors find that existing approaches perform worse than the directly fine-tuning when the source and target domains are apart. In order to tackling such challenging setting, the proposed method leverages Fisher informance to measure the importance on the target domain and performs the corresponding modulation. Experimental results demonstrate the effectiveness of the proposed approache, especially across the relatively challenging domains: \eg from human faces to animal faces.

**Questions:**

Major concerns:
- As CDC evaluates its approach on skteches as well, I am wondering how well the proposed method performs under such drawing style changes. Ideally, there might be only several kernels affecting the drawing style when adapting from face to sketch faces. Thus, evaluating the proposed method under such style changes is very necessary.
-  How much the proximity between the source and the target could be relaxed. As the main focus is to challenge the domains that are apart to some extent, what if we fine-tune a face model on other objects like cars? Are all kernels useless? That is, the limit across challenging domains of the proposed method is worth to explore.
- How many images of target domains are required to determine the importance of kernels. As all experiments are conducted with 10-shot samples, what if we continue to decrease the number of training samples, like 5, even 1 image? And what the FID becomes if we continue to involve more target images, (maybe involving all AFHQ) as Fig.6 does (but more than 200 images).
- The technical contributions could be better clarified, compared to EWC. It seems that this paper measures the importance on the target domain while EWC does it on the source domain.

Overall, the exploration on such challenging domains is very necessary for the future research of few-shot generation. Thus, I will change my score depending on the response.





**Limitations:**

The limitations of the proposed method could be discussed more comprehensively.

**Strengths And Weaknesses:**

- This paper is well-organized and easy to follow.
- The focus on relatively unrelated domains is very interesting and challenging as existing methods just fail under this.
- Anlysis of evaluating previous approaches is also well-built to uncover the issues.
- Experimental results demonstrate the effectiveness of the proposed adaptation method that explicitly considers the target domain.

---

> ### Author Response · Authors · 2022-08-02
> **[Response for Reviewer ekht] Part 2/2**
>
> $ $
>
> > As CDC evaluates its approach on sketches as well, I am wondering how well the proposed method performs under such drawing style changes. Ideally, there might be only several kernels affecting the drawing style when adapting from face to sketch faces. Thus, evaluating the proposed method under such style changes is very necessary.
>
> We thank the Reviewer for the comment. We would like to point out that evaluation based on the Sketches dataset has already been included in the Supplementary (Sec E.2, Figure S12). We will include a pointer in the main paper to make this clear. As shown in Figure S12, our generated images have good quality and diversity. In addition to our submission, here we further evaluate our method with other baseline and SOTA works on the Sketch domain: https://drive.google.com/drive/folders/1TqcKRcSzrOPybybWkRNF3yQvtsrf0ofu?usp=sharing
>
> As shown in our results, our method is competitive for Sketches dataset.
>
>
> $ $
>
> > How many images of target domains are required to determine the importance of kernels. As all experiments are conducted with 10-shot samples, what if we continue to decrease the number of training samples, like 5, even 1 image? And what the FID becomes if we continue to involve more target images, (maybe involving all AFHQ) as Fig.6 does (but more than 200 images).
>
> We thank the Reviewer for the comment. We follow the Reviewer’s suggestion and extend Figure 6 with more data points (1-shot, 5-shot,  500-shot, 1k-shot, and the entire AFHQ-Cat). We perform adaptation on 256$\times$256 resolution.
>
> The results can be found here: https://drive.google.com/drive/folders/1V94eJ428-4J4dI-IchfjF6a3b14dUwRD?usp=sharing.
> As shown in the Figure, our proposed method is very competitive in the low data regime and comparable to other methods when the whole AFHQ-Cat data is available.
>
>
> $ $
>
> > The technical contributions could be better clarified, compared to EWC. It seems that this paper measures the importance on the target domain while EWC does it on the source domain.
>
> We thank the Reviewer for the comment. As pointed out correctly by the Reviewer, one of the main novelties of our method compared to existing work (including EWC) is that we propose a **target-aware** FSIG approach; in particular, we determine the importance of kernels w.r.t. target domain for knowledge selection using our proposed importance probing method.
>
> In addition, we propose to adaptively preserve useful knowledge via parameter-efficient **rank-constrained kernel modulation** (**KML**; see Ln 201 to Ln 230 in main paper), which is significantly different compared to EWC and other existing work. In the Supplementary (Sec C, Table S2), we show that our method outperforms not only EWC but also a target-aware version of EWC (EWC + importance probing), highlighting the improvement due to the ranked-constrained KML.
>
> Furthermore, in Sec 3 (and Supplementary Figure S2), we **analyze existing SOTA via the lens of source-target domain proximity**. Our systematic and comprehensive analysis reveals a critical research gap in existing work: existing SOTA based on target-agnostic approaches are inadequate to handle general few-shot image generation where the source and target domains are more apart. We believe our analysis and our findings are important contributions to few-shot image generation.
>
> $ $
>
> > The limitations of the proposed method could be discussed more comprehensively.
>
> We thank the Reviewer for the suggestion. We will discuss the limitations of the proposed method more comprehensively. In particular, as discussed in our response to Qn 2, our method still has room for improvement in the very challenging setups when adapting between two very distant domains such as FFHQ$\rightarrow$Cars. We will include such results to improve the discussion. We remark that, on one hand, better ideas that can achieve more accurate important kernel identification and important knowledge preservation may improve the results in such very challenging setups. On the other hand, there may not be much knowledge in the source domain that can be generalized to the target domain when the domains are very distant. The Reviewer’s fundamental question *“How much the proximity between the source and the target could be relaxed”* is an important one to be studied in few-shot image generation.

---

> ### Author Response · Authors · 2022-08-02
> **[Response for Reviewer ekht] Part 1/2**
>
> We sincerely appreciate Reviewer’s positive feedback on our work. Based on comments from Reviewer, we will add more analysis for Sketches, more results for challenging domains (FFHQ$\rightarrow$Cars), and *expand* the performance analysis under the various number of shots. We hope these details and experimental results address the concerns of the Reviewer.
>
> $ $
>
> > How much the proximity between the source and the target could be relaxed. As the main focus is to challenge the domains that are apart to some extent, what if we fine-tune a face model on other objects like cars? Are all kernels useless? That is, the limit across challenging domains of the proposed method is worth to explore.
>
> Thank you for the comment. Reviewer’s question *“How much the proximity between the source and the target could be relaxed”* is an important and fundamental one for the area of few-shot image generation. While a comprehensive and in-depth study may be required to completely address the Reviewer’s question, here we provide some responses to this. Furthermore, we provide experiment results suggested by the Reviewer. We appreciate feedback and comments from the Reviewer on our response.
>
> First, we remark that the *upper bound* on proximity between source domain S and target domain T (refer to as “proximity bound” in our response and conceptually the same as the Reviewer’s notion of *“the limit across challenging domains”*) could be conditioning on (a) number of available samples (shots) from target domain, and (b) method used for knowledge transfer.
>
> * **(a) Proximity bound conditioning on the number of target domain samples.** In this paper, we focus on few-shot setups, e.g. 10 shots. However, with more target domain samples available, proximity between S and T can be further relaxed, and the proximity bound would increase, i.e. for a given generative model on S, we could learn an adapted model for T which is more distant. Intuitively, increasing the number of target domain samples can provide more diverse knowledge for T, and as a result, there is less reliance on the knowledge of S that is generalizable for T (which would decrease as S and T are more apart). In the limiting cases when abundant target domain samples are available, knowledge of S would not be critical, and proximity constraints between S and T may be totally relaxed (ignored). We will discuss the effect of the number of shots when answering the Reviewer's next question.
>
> * **(b) Proximity bound conditioning on the knowledge transfer method.** Given a generative model pretrained on S and a certain number of available samples from T, the method used for knowledge transfer plays a critical role. If the method is superior in identifying **suitable** transferable knowledge from S to T, the proximity between S and T can be relaxed, and the proximity bound would increase. In our work, our first contribution is to reveal that existing SOTA approaches (which are based on target-agnostic ideas) are **inadequate** in identifying transferable knowledge from S to T. As a result, when proximity between S and T is relaxed, the performance of adapted models is miserably poor, as discussed in Sec 3, Sec 5, and Appendix. Therefore, our second contribution is to propose a target-aware approach that could identify more meaningful transferable knowledge from S to T, allowing relaxation of the proximity constraint.
>
> We remark that, unlike knowledge transfer in discriminative models [20][44][24][27], problem of knowledge transfer in generative models (esp. under few shots) has only been sparsely studied e.g., [3][4]. We believe our work makes an important step and highlights some important subtleties in the more general setups when S and T could be apart.
>
> In addition, based on the Reviewer’s suggestion, we provide experimental results for adaptation between two very distant domains: FFHQ$\rightarrow$Cars using only 10-shots, aiming to answer two main questions: (1) the availability of transferable knowledge from FFHQ to Cars, and (2) Performance of proposed method in this setup. The results are as follows:
>
> |Method|FID($\downarrow$)|Intra-LPIPS($\uparrow$)|
> | ------------- |:-------------:|:-------------:|
> |Training from Scratch|47.67|0.649|
> |EWC|228.35|**0.716**|
> |CDC|209.07|0.640|
> |DCL|229.32|0.685|
> |Ours|**36.06**|0.685|
>
> Results show there is still useful and transferable knowledge from FFHQ$\rightarrow$Cars (e.g. low-level edges, shapes), leading to better performance in the adapted model using proposed method compared to training from scratch. In addition, our proposed method can identify and transfer more meaningful knowledge compared to other methods, resulting in lower FID and higher diversity in generated images.
>
> Some generated samples can be found in this anonymous link:
> https://drive.google.com/drive/folders/1a_ENA1wcqBl5vKrD_MfqVODktmIJKbt6?usp=sharing. The visual results show that our method can produce more reasonable images compared to other methods.

---

> ### Author Response · Authors · 2022-08-08
> **We thank the valuable time from Reviewer ekht**
>
> Dear Reviewer **ekht**:
>
> We sincerely thank you for your valuable time.
>
> In our previous responses, we have tried our best to address all questions by Reviewer **ekht**. We would like to ask if our responses are adequate. If not, we would be happy to provide more information.
>
> Moreover, we would like to thank Reviewer **ekht**’s suggestion of **experiment FFHQ $\rightarrow$ Cars (FSIG with remote target domain), which further strengthens the main findings in our paper:**
>
> - SOTA FSIG methods including EWC, CDC, DCL perform no better than simple approaches such as fine-tuning and training from scratch when the proximity assumption between source and target domains is relaxed. The results of FFHQ $\rightarrow$ Cars in our previous response are consistent with our findings in Sec. 3, Sec. 5 and Supplementary.
>
> - Our proposed Adaptation-aware approach improves the performance of the adapted model and performs better than all baseline and SOTA methods.
>
> **For all experiments in the submission and rebuttal, we provide the Pytorch code/checkpoints for reproducible research**.
>
> Additionally, we thank Reviewer **ekht** for the very positive score especially the **“excellent”** rating for “Soundness” and “Presentation”, and **“good”** for “Contribution”:
>
> - **Soundness**: 4 excellent
> - **Presentation**: 4 excellent
> - **Contribution**: 3 good
> - Rating: 5: Borderline accept
>
> Given the positive assessment and rating, with our additional response and further results, **we would like to humbly request Reviewer ekht to consider increasing the overall “Rating” if there is no further question**.
>
> We thank you for your valuable time and consideration.

---

> ### Author Response · Authors · 2022-08-09
> **We thank Reviewer ekht**
>
> Dear Reviewer **ekht**,
>
> As today is the last day of the author-reviewer discussion phase, we would like to ask if our responses are adequate. If not, we would be happy to provide more information.
>
> Thank you for your time.
>
>
> Best,
>
> Authors

---

> > ### Comment · Reviewer_ekht · 2022-08-09
> > **Thanks for the response**
> >
> > Thanks for the efforts and the detailed response which addressed my concerns. Thus I will raise my rating to 6 weak accept.

---

### Author Response · Authors · 2022-08-02
**Thank You for the Positive and Constructive Comments**

We thank all Reviewers for the positive and constructive comments. We believe our work is an important step in addressing **general** few-shot image generation (FSIG):

* As our first contribution, we reveal, for the first time, a critical **research gap** in all existing state-of-the-art FSIG approaches: they fail to consider the target domain in selecting the source model’s knowledge to be preserved in the adapted model. We conduct a comprehensive empirical analysis to reveal that these **target-agnostic** approaches in all existing state-of-the-art could be **problematic**: when close proximity assumption between source and target domains is relaxed, the performance of existing state-of-the-art is miserably poor, generating samples that are inconsistent and incompatible with the target distribution (e.g., samples of “Cat with safety helmet”, see main paper Figure 3). Comprehensive analysis can be found in the main paper Sec 3, Sec 5, and Appendix.

* Informed by our analysis, as our second contribution, we propose a novel **target domain-aware** approach to judiciously select the source model’s knowledge into the adapted model. Our approach includes the **importance probing** algorithm and a parameter-efficient **rank-constrained kernel modulation** to identify/select “important” kernels – kernels that encode meaningful knowledge for adaptation to the target domain. We conduct extensive experiments to show that our proposed method consistently achieves SOTA performance across source/target domains of different proximity, including challenging setups when source/target domains are more apart. The proposed approach is discussed in Sec 4, extensive comparisons are included in Sec 5 and Appendix.

Our submission includes program code and details for reproducible research.

In what follows, we provide comprehensive responses to all questions. For the responses that require figures/images, we have used anonymized links to not violate the double-blind reviewing policy of NeurIPS. We could provide more details if there are further questions. We hope that our responses can address the concerns and we sincerely hope that reviewers may consider increasing the ratings if our responses are satisfactory.

---

### Author Response · Authors · 2022-08-05
**Thanks again for your constructive review and comments on our submission.**

Dear AC, Reviewers:

We sincerely thank you for your valuable time.

**In our previous posts**, we have addressed each and every concern of all our Reviewers in separate responses with **7 sets of additional requested experiments and analysis**. All these additional results and analysis further strengthen our proposed FSIG method.

**In this post**, we would like to let you know that we have integrated these extensive additional results and analysis into our paper. The revised Supplementary version includes:

- **Section D.2**: Additional GAN Architectures (Reviewer **4Rin**)

- **Section D.3**: Alternative characterization of importance measure using Class Saliency (Reviewer **gbAJ**)

- **Section D.4**: Comparison with Adaptive Data Augmentation (Reviewer **gbAJ**)

- **Section D.5**: Importance probing with an extremely limited number of samples (Reviewer **ekht**)

- **Section E**: Discussion: What form of visual information is encoded by high FI kernels? (Reviewer **4Rin**)

- **Section G**: Discussion: How much the proximity between the source and the target could be relaxed? (Reviewer **ekht**)

- **Section H.1**: Updated Potential Societal Impact (Reviewer **gbAJ**)

--

All the code/models for our experiments (including the additional one used in our previous posts) can be found here (please refer to the readme within the folder for more details):

`https://drive.google.com/drive/folders/1aVfKnUIRKmFHODGeF4vvcNeMZC7H5E8A?usp=sharing`

We hope that we have addressed the Reviewers’ questions satisfactorily and look forward to feedback from our Reviewers.

We sincerely appreciate your valuable time.

---

### Author Response · Authors · 2022-08-09
**We thank ACs and Reviewers.**

Dear ACs and  Reviewers:

We sincerely thank you for your positive comments and recommendations. We appreciate our Reviewers’ involvement during the discussion stage allowing us to accurately communicate additional details of our work.

-------------------------------------

We highly appreciate the Reviewers’ positive feedback:

=========== Reviewer **ekht** ===========

- Soundness: **4 excellent**
- Presentation: **4 excellent**
- Contribution: **3 good**

Rating: 6: Weak Accept: Technically solid, moderate-to-high impact paper, with no major concerns with respect to evaluation, resources, reproducibility, ethical considerations.


=========== Reviewer **4Rin** ===========

- Soundness: **4 excellent**
- Presentation: **3 good**
- Contribution: **3 good**

Rating: 7: Accept: Technically solid paper, with high impact on at least one sub-area, or moderate-to-high impact on more than one areas, with good-to-excellent evaluation, resources, reproducibility, and no unaddressed ethical considerations.


=========== Reviewer **gbAJ** ===========

- Soundness: **3 good**
- Presentation: **3 good**
- Contribution: **3 good**

Rating: 6: Weak Accept: Technically solid, moderate-to-high impact paper, with no major concerns with respect to evaluation, resources, reproducibility, ethical considerations.

-----------------------------------
For all experiments, our Pytorch code/models are available (anonymous link) for reproducible research.

`https://drive.google.com/drive/folders/1aVfKnUIRKmFHODGeF4vvcNeMZC7H5E8A?usp=sharing`

We sincerely thank you for your valuable time.

Best,

Authors

---

### Meta-Review · Area_Chair_9uWh · 2022-09-05

**Recommendation:** Accept
**Confidence:** Certain

**Metareview:**

This paper focuses on the few-shot image generation task, which is an interesting and challenging problem. It is well-organized and easy to follow. All the reviewers acknowledge that experimental results demonstrate the effectiveness of the proposed adaptation-aware kernel modulation method. Overall, the meta-reviewer recommends acceptance of the paper.

**Award:**

No

---

### Decision · Program_Chairs · 2022-09-14

Accept